# Frequency stable and low phase noise THz synthesis for precision spectroscopy

Léo Djevahirdjian [1] ✉, Loïc Lechevallier[1] ✉, Marie-Aline Martin-Drumel [2] ✉, Olivier Pirali[2] ✉, Guillaume Ducournau [3] ✉, Rédha Kassi[3] ✉ & Samir Kassi [1] ✉

We present a robust approach to generate a continuously tunable, low phase noise, Hz linewidth and mHz/s stability THz emission in the 0.1 THz to 1.4 THz range. This is achieved by photomixing two commercial telecom, distributed feedback lasers locked by optical-feedback onto a single highly stable V-shaped optical cavity. The phase noise is evaluated up to 1.2 THz, demonstrating Hz-level linewidth. To illustrate the spectral performances and agility of the source, low pressure absorption lines of methanol and water vapors have been recorded up to 1.4 THz. In addition, the hyperfine structure of a water line at 556.9 GHz, obtained by saturation spectroscopy, is also reported, resolving spectral features displaying a full-width at half-maximum of 10 kHz. The present results unambiguously establish the performances of this source for ultra-high resolution molecular physics.

In gas phase molecular physics, the terahertz (THz) range offers new possibilities for accurate characterization of the rotational structure of light molecules[1] as well as the low-lying vibrational modes of heavier ones[2]. Such measurements are especially relevant for astrophysics[3,4], trace compound detection[5] and healthcare applications[6]. THz technology is also the root of free space telecommunications and the future of high data-rate telecommunications[7]. All these fields require high spectral purity, thus low phase noise, together with easy tuning and wide frequency range coverage. Very stable and sub-kHz sources are required for high-resolution spectroscopy[8,9], while very low phase noise is a key point for achieving ultra-high data-rates[10] for telecoms. Several reviews, e.g. refs. 11,12, report the advantages and limitations of various instrumental approaches for THz generation and spectroscopy.

In particular, frequency multiplication chains, widely used for molecular spectroscopy, can continuously cover a series of spectral windows up to 1.5 THz. Above this frequency, up to 2.75 THz, only narrow spectral windows—typically a few percent of the central frequency (i.e. hundreds of GHz at THz frequencies)—can be reached with high-frequency purity[13] but by using expensive laboratory equipment

with limited output power. This spectrally segmented approach forces users to add or remove elements based on the spectral window they wish to use, which is time-consuming and requires optical re-alignments. Because amplitude and phase modulations are essentially possible prior to frequency multiplication, which strongly limits the achievable communications bandwidth, this approach cannot be efficiently used in the telecommunications field[10,14]. From the optical side, THz quantum cascade lasers (THz-QCLs)[15] lase above 1 THz with high power and potentially narrow linewidth but are only available in specialized labs and only cover limited spectral ranges. They remain difficult to operate in the context of ultra-high-resolution spectroscopy and do not allow for phase and amplitude modulation. Recently, by replacing traditional $CO_2$ laser optical pumping with mid-infrared QCLs, molecular lasers have benefited from a renewed interest as they now offer numerous THz laser lines[16–18]. But they still suffer from both tuning limitations, that make them relatively difficult to use in the context of broadband molecular spectroscopy, and the impossibility of phase tuning that makes them inadequate for encoding information. Recent literature studies have shown that Difference Frequency Generation (DFG) using a pair of lasers is an extremely promising

[1]Univ. Grenoble Alpes, CNRS, LIPhy, Grenoble, France. [2]Université Paris-Saclay, CNRS, Institut des Sciences Moléculaires d'Orsay, Orsay, France. [3]Université de Lille, CNRS, Centrale Lille, Univ. Polytechnique Hauts-de-France, UMR 8520 IEMN, Institut d'Electronique de Microélectronique et de Nanotechnologie, 59655 Villeneuve d'Ascq, France. ✉e-mail: leo.djevahirdjian@univ-grenoble-alpes.fr; loic.lechevallier@univ-grenoble-alpes.fr; marie-aline.martin@universite-paris-saclay.fr; olivier.pirali@universite-paris-saclay.fr; guillaume.ducournau@univ-lille.fr; redha.kassi@univ-lille.fr; samir.kassi@univ-grenoble-alpes.fr

approach for synthesizing THz frequencies while combining spectral resolution and tunability (e.g. ref. 12). Moreover, by literally transposing the telecom band into the THz band, this approach directly benefits from the unmatched performances of technologically mature Near-Infrared (NIR) electro-optical modulators.

The DFG technique consists of photomixing[19] two independent NIR lasers in a medium equipped with an antenna that emits photons at the difference frequency of the lasers. The emission linewidth is then the quadratic sum of the frequency noise of each laser, which is in the MHz range[20]. In the literature, DFG in the THz (DFG-THz) has been demonstrated by locking a pair of external cavity diode lasers (ECDL) to an optical frequency comb (OFC) around 800[21] and 1550 nm[22] allowing for absolute frequency determination and resolution as good as 10 kHz over 3 THz[22]. DFG was also demonstrated using broadly available distributed feedback DFB lasers (DFB) in the 1550 nm telecom range[23]. It has already been applied to spectroscopy[21] and telecommunications[20,24,25] and considerable efforts have been made to narrow the THz effective linewidth down to the tens of kHz[22,26–30], and very recently sub-Hz level[31], often at the price of complex setups. In this work, we demonstrate that the optical-feedback locking of two DFB lasers onto the resonance frequency modes of a single cavity is a straightforward and robust way to obtain very low phase noise difference frequencies. Optical feedback makes the laser frequency narrowing easier, leading to emission frequencies that clone the cavity stability. Moreover, since the main noise of the optical cavity comes from fluctuations in its length, the common mode noise of the lasers largely cancels out in the frequency difference operation[27].

## Results

### Experimental setup

The THz frequency generation scheme employed in this study is depicted in Fig. 1. It is based on the photomixing of two telecoms DFB lasers, both lasing in the NIR, around 190 THz (1.5 μm wavelength), with about 20 mW of power each. The telecom grade DFBs present the advantage of fast and mode hop-free temperature tuning of the lasing frequency, typically over one THz. By tuning both DFBs, a frequency difference of up to 2 THz and beyond, by switching additional lasers, can easily be obtained[32]. In the free-running regime, each DFB exhibits

MHz level linewidths with no optical phase relation between them. By quadratic summation of both contributions, this translates to an even wider THz photomixed linewidth, as observed in commercial systems. The frequency resolution obtained with such a source is obviously inadequate for tackling low-pressure and high-precision gas phase spectroscopy, for which the molecular linewidth is typically sub-MHz at room temperature, or phase-sensitive applications such as dense telecom encoding. Actively stabilizing DFB lasers against an optical cavity resonance using laser diode current feedback[30], partial[33] or full[34,35] feed-forward corrections, remains a challenging task necessitating a high bandwidth feedback loop, frequency control, or an OFC. As an alternative, we propose an approach that circumvents these bandwidth and frequency issues by exploiting optical feedback[36] as a robust locking mechanism of the lasers onto a single very stable high finesse optical cavity. For the present exploratory study, we used the previously described very stable three-mirror V-shaped resonator as a reference optical cavity[37]. This geometry avoids unwanted direct back-reflection when the laser is injected through the folding mirror while only resonant photons of the optical cavity propagate back to the laser[38]. The physical mechanism of optical feedback (OF) between the laser and the optical cavity modes induces a drastic reduction in the emission linewidth, typically by six orders of magnitude from MHz to Hz level, well below the kHz-level cavity mode width[36,39]. The very stable optical cavity we used has a finesse of 300,000 and about 491 MHz free spectral range (FSR = $c/(2L_1 + 2L_2)$, where $L_1$ and $L_2$ are the two cavity arm lengths). It is made of a vertically mounted 15 cm long ultra-low-expansion glass (ULE) spacer tube in contact with two invar mirror holders maintained with springs on each side. The assembly is kept in an ultra-high vacuum and is temperature stabilized at 294 K, with 10 mK stability, leading to optical frequency drift close to 10 Hz/s at 193 THz (1.55 μm wavelength). This simplified design is certainly not as powerful as the state-of-the-art optical cavities used for optical clocks, but it is perfectly suited to a wide range of applications, particularly gas-phase spectroscopy. For a detailed description of the cavity and the OF mechanism, we refer the reader to ref. 37 and the references therein.

The setup, illustrated in Fig. 1, mainly relies on commercially available polarization maintaining (PM), fiber-based telecom

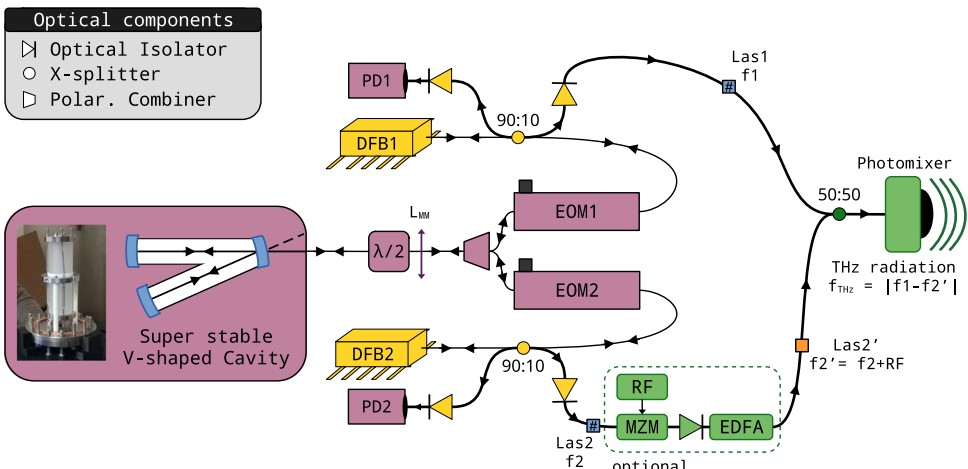

**Fig. 1 | Experimental setup.** Schematic of the THz source setup based on locking two distributed feedback lasers (DFB1, DFB2) against the $TEM_{00}$ modes of a single V-shaped optical cavity. The polarization-maintaining fiber optical path from lasers to primary outputs (Las1, Las2) are identical. The laser's polarization axis is aligned to the optical fiber slow axis and FC/APC connectors key. Each line includes a 90:10 fiber splitter and an electro-optic modulator (EOM). A fibered polarization combiner permits the polarization of the lasers to be crossed before they are injected into the cavity. A series of optical isolators protect the system from back-reflection from the monitoring photodiodes (PD1, PD2) and from the output lines (Las1,

Las2). As an option, an RF generator (RF) and a Mach-Zehnder (MZM) I/Q modulator configured as a 2–20 GHz single sideband generator with 30 dB contrast allow instantaneous, fine and arbitrary tuning of Las2 optical frequency. An optical amplifier (EDFA) compensates for the MZM insertion losses. The two lasers, which have the same polarization, are combined using a 50:50 power-splitter and injected into a photomixer that emits at their difference frequency. A coarse THz tuning, from 0 to 1.7 THz, is obtained in a few seconds by adjusting the temperature of the DFB1 and/or DFB2 chips.

components. To achieve efficient OF, the commercial DFB lasers (DFB1 and DFB2) have no internal optical isolator. Two independent optical lines with the polarization aligned on the slow axis are combined through a fiber polarization combiner, which permits the polarization of the lasers to be crossed orthogonally before they are coupled in the V-shaped optical cavity by means of a single lens and a pair of steering mirrors. A free-space, half-wave plate provides precise matching with the cavity modes polarization and avoids cavity-induced laser cross-talk. Interestingly, the transmission of the $TEM_{00}$ modes of the cavity−which propagates back to the laser−are naturally re-coupled to the fiber, both in space and polarization. The lasers are therefore highly independent of each other.

In more detail, each laser is injected into a 90:10 X-splitter, which extracts 90% of the power for applications. The remaining 10% are passing through an Electro-Optic phase Modulator (EOM). The back-propagating cavity transmission is captured with a photodiode installed on the 90% complementary port of the X-splitter while the 10% remaining power is directed to the laser for optical feedback. To control the phase between the lasers and the cavity, individual DC voltages are applied to each EOM. They are evaluated from the transmission of the cavity using a first derivative scheme. To do this, a weak modulation is summed to the EOM voltage (19 and 20 kHz to DFB1 and DFB2 lines, respectively), which induces a cavity transmission modulation whose relative phase is representative of the laser and cavity phase mismatch. These derivative-type error signals, retrieved using lock-in amplifiers, are kept to zero using a low bandwidth (<1 kHz) proportional-integral feedback loop acting on the EOM DC voltage components. The optical feedback regime can be maintained over hours in a single mode. However, if the laser current or temperature is strongly changed, the optical feedback mechanism−which works over a limited locking range controlled by the optical feedback level[36]−is interrupted. A relocking then occurs on the next cavity mode, providing efficient mode-by-mode frequency hopping[38]. Finally, the two independent output lines having the same output polarization are mixed together in a 50:50 X-combiner. A set of optical isolators prevents unwanted optical feedback from the user and photodiode ports. Given that the DFB1 and DFB2 wavelengths can be tuned from 1543 to 1550 and 1550 to 1557 nm, respectively, the difference frequency can be coarsely tuned from 0 to 1.7 THz with a step resolution of one cavity FSR.

Theoretically, optical feedback allows for sub-Hz level linewidth reduction[36] and both lasers inherit the stability properties of the reference cavity. But the latter suffers from mechanical vibration or thermo-mechanical constraints that affect the cavity geometry and thus the cavity FSR. Typically sub-kHz linewidth emission in the NIR domain is obtained with the cavity used in the present study[37]. However, because both lasers are locked against the same optical cavity, the noise partially cancels with the difference frequency process within the ratio of the optical and THz frequencies. Since this ratio is close to 200, Hz-level linewidth of the emission is expected in the THz domain. Recent experimental results[40] and modelings[41] let us envisage sub-Hz emission in the near future using a better-designed optical resonator.

Optionally, to precisely and continuously tune the THz frequency, a Mach−Zehnder Modulator (MZM, here a MXIQER-LN-30 from iXBLUE) configured for optical single side-band generation (OSSBM) with 30 dB contrast, was inserted into the DFB2 line. Its 10 dB insertion losses are compensated with an optical amplifier (BOA, Thorlabs). The MZM provides fine frequency tuning by adding or subtracting a Radio-Frequency (RF in the 2−20 GHz range) to/from the optical frequency. This optical shift is directly reflected in the down-mixed signal with a 1:1 scaling.

## Source spectral performances
In order to characterize the phase noise and long-term frequency stability of our DFG source, we performed a series of supplementary measurements. These characterization setups are schematically represented in Fig. 2 and the results are reported hereafter.

## Phase noise in the RF domain
To quantitatively characterize the short-term frequency noise of the source, the difference of the two laser frequencies was set to 1.965 GHz (4 FSR) and photomixed in a fast photodiode (TDET08CFC/M, Thorlabs), see panel (a) of Fig. 2. The beating was down-converted to 100 kHz by mixing with a GPS time referenced synthesizer (Rohde & Schwarz SMB-100A) tuned to 1.9650001 GHz. The resulting signal was sampled at 1 MS/s with a fast 16-bit acquisition card (PCIe6251, National Instruments) over 1 s. The power spectral density (PSD), shown on panel (a) of Fig. 3, was then computed and normalized to the peak maximum. The PSD curve indicates that the full-width at half-maximum (FWHM, −3 dB/Hz on PSD) beatnote arrives at the Hz level, which is limited by the 1 s Fourier transform resolution. The beating pedestal observed below −20 dB is expected to have three combined origins: (1) the lock-in mechanism and its electronics, (2) cavity instabilities, and (3) the noise floor of the down-mixing chain.

A phase noise spectrum was recorded using a spectrum analyzer (Rohde & Schwarz FSP 9 kHz to 13 GHz equipped with the FS-K40 option). It corresponds to the purple trace in panel (b) of Fig. 3 and must be compared to the reference trace (in green) obtained with a GPS-referenced synthesizer tuned to the same frequency and set to the same power. The spectrum analyzer noise floor, as given by the instrument data-sheet[42], is also indicated above 100 Hz (red dots). The phase noise spectrum starts with −20 dBz/Hz at 1 Hz and decreases with a slope of −20 dBc/decade up to 50 Hz, a typical behavior for electronic noise. It is then followed by a steeper slope of about −40 dBc/decade up to 1 kHz. This is typical of a $1/f^2$ noise process linked to a random FM modulation which may arise from the active locking system that generates voltage fluctuations on the EOMs. Above 300 Hz the DFG arrives at the instrumental noise floor confirming that the OF mechanism fully controls the DFB emission.

## Long-term stability
The long-term stability of our setup has been characterized by monitoring over 37.5 h both the difference frequency, set to 1.965 GHz, and the absolute NIR frequency of Las1. The laser frequency was determined by recording its beatnote signal with a GPS-referenced OFC (FC1500, Menlo Systems), see panel (a) of Fig. 2. The beating signal was measured every second with 1 Hz resolution using a fast acquisition card. The laser frequency and the difference frequency drifts are plotted in panels (a) and (b) of Fig. 4, respectively.

An overlapping Allan deviation[43] of the difference frequency has been computed with a Python script based on the *allantool*[44] library and is shown in panel (c) of Fig. 4. It confirms that the difference frequency linewidth is lower than 1 Hz for a single acquisition. Over 37.5 h, an averaged frequency drift of 0.15 mHz/s is found as revealed by the trend of the blue trace in panel (b).

The long-term cavity FSR drift affects both NIR lasers (Las1 and Las2) with coefficients $i$ and $j$ which are the corresponding cavity longitudinal mode numbers. Because the mode numbers are different, the difference frequency is affected by the ratio $(i–j)/i$. For the present measurement, this ratio is $1.015 \times 10^{-5}$. In Fig. 4 we report the difference frequency shift (blue curve) together with the absolute frequency shift, scaled with the above factor (orange curve). The very good agreement of the trend of the two curves indicates that the slow difference-frequency drift arises from the cavity FSR fluctuation. However, short-term excursions of about ±10 Hz are clearly visible. They mainly result from slowly varying optical etaloning due to the fiber connectors, which biases the first-derivative error signal that feeds the EOM control electronics.

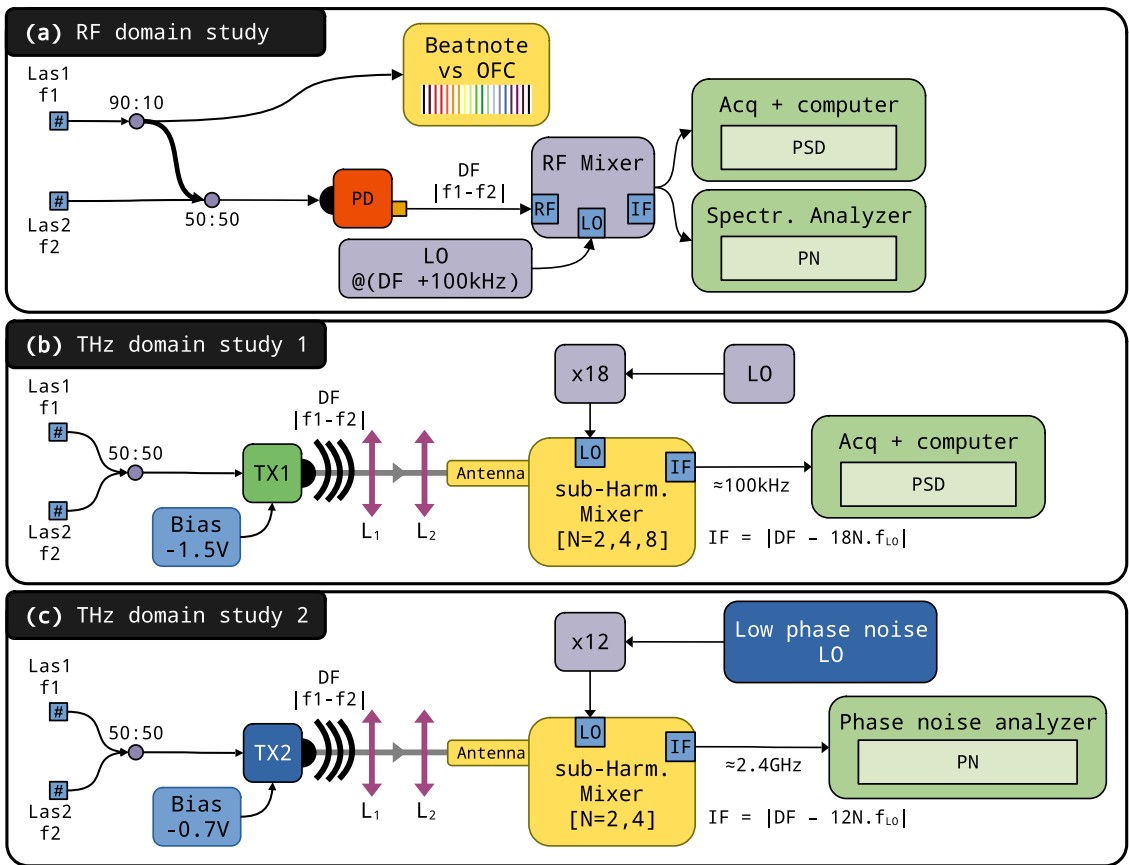

**Fig. 2 | Experimental setups used to characterize the spectral performances of the source. a** 10% of the Las1 optical power is extracted and mixed with the output of a GPS-referenced OFC to monitor the long-term frequency stability of Las1. The remaining power is mixed with Las2 on a common PM fiber. The frequency difference (DF) beating is detected with a fast-photodiode (PD) and down-mixed to about 100 kHz using an RF Mixer driven with a local oscillator (LO) set 100 kHz above DF. The power spectral density (PSD) and phase noise (PN) of the down-converted signal are evaluated using an FFT-based procedure and the PN function of a spectrum analyser, respectively. **b** Setup used for PSD characterization at THz frequencies up to 1.2 THz using a commercial photomixer (TX1). The signal is down-converted in the 100 kHz range using a sub-harmonic mixer (SHM) driven by a Schottky-diode-based frequency multiplication chain fed by a 12.5 GHz local oscillator. Total multiplication factors *M* were set between 24 and 96. The THz emission was focused in the SHM horn-antenna using two lenses (L₁), (L₂) with 50 and 100 mm focal lengths. The PSD of the resulting signal was analyzed using the FFT-based procedure. **c** Setup used for the PN characterization of the THz emission at 300 and 600 GHz. The difference frequency was generated with an optimized photomixer (TX2). Two-lenses (L₁), (L₂) with 50 mm focal lengths were used to efficiently inject the SHM horn antenna, while it was pumped by a state-of-the-art electronic chain. The PN of the resulting beating signal, set around 2.4 GHz, was characterized using a high-performance phase noise analyser.

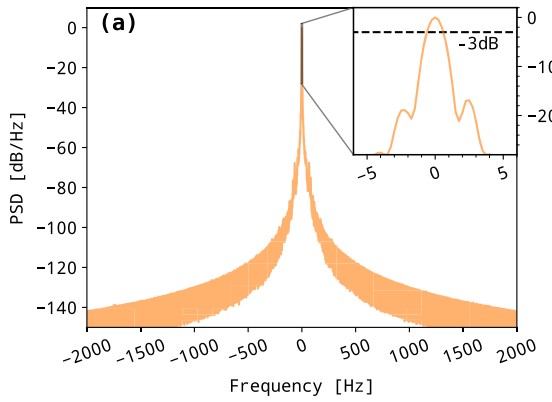

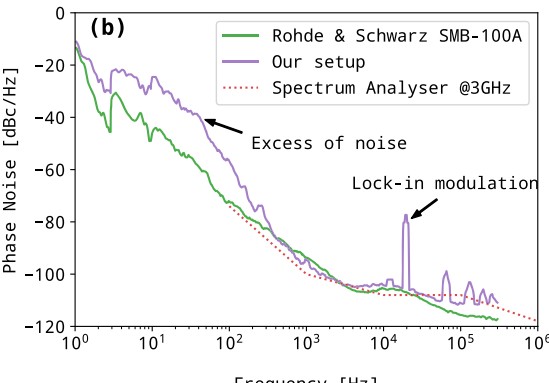

**Fig. 3 | Source noise characterisation in the RF domain. a** Power Spectral Density for a laser difference frequency of 1.965 GHz (4 cavity FSR) recorded by the acquisition card after down-frequency mixing to 100 kHz. The central frequency is numerically shifted to 0 for readability and the PSD is normalized to peak maximum. On the top right, a zoom on the central region, the dotted line indicating the width at −3 dB/Hz, i.e., the FWHM. **b** Phase noise spectrum for a 1.965 GHz difference frequency (purple line) compared to the phase noise observed for a similar frequency generated using an RF synthesizer (green line) together with the spectrum analyzer phase noise specification at 3 GHz (red dots).

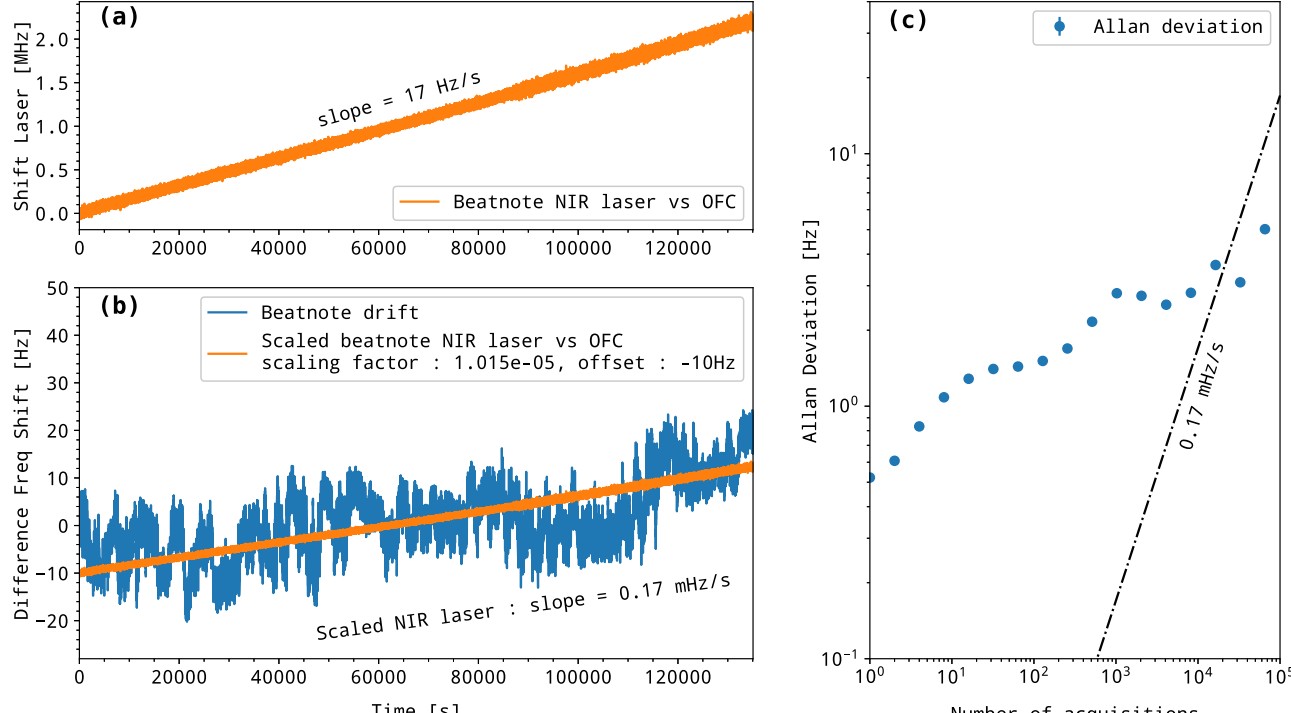

**Fig. 4 | Long-term stability characterisation of a 1.965 GHz beatnote generated by the DFG setup. a** Raw measurement of the Las1 optical frequency drift. **b** Experimental difference frequency beatnote drift (blue). Calculated beatnote drift (orange trace) applying a scaling factor of $1.015 \times 10^{-5}$ to the Las1 drift plotted in panel **a**, this trace is offset by 10 Hz for better readability. See text for details. **c** Overlapping Allan deviation plot of the experimental difference frequency data from panel **b** showing sub-Hz level stability on short time and 0.17 mHz/s long-term drift.

## Performance in the THz domain

Following the characterization of the performances of our DFG in the RF domain, we extended our characterization for frequencies generated in the THz range using the setup described in panel (b) of Fig. 2. Scaling up the frequency of the generated beatnote to the THz range is easily done by adjusting the temperature of the lasers in order to lock them to more distant cavity modes. To produce the THz emission, the lasers are mixed in a commercial photomixer (PCA-FD-1550-100-TX-1, Toptica). The spectral characterization of the THz source was performed using heterodyne detection, by focusing the THz radiation in a sub-harmonic mixer (WR2.8SHM, VDI) equipped with a horn antenna (WR3.4DH and WR2.2H, VDI for the 220–330 and 330–500 GHz ranges, respectively). The sub-harmonic mixer (SHM) was seeded by a GPS-referenced synthesizer (Rohde & Schwarz SMF-100A with low phase noise option) feeding a 36-fold multiplication chain. The local oscillator frequency was chosen to obtain a beating frequency at the mixer output close to 100 kHz. This beating signal was amplified (ZFL500-LN+, Minicircuits) and recorded using an acquisition board. This signal provided information on the relative frequency (or phase) noise between the two THz sources.

The frequency of the synthesizer was adjusted to produce radiation in the frequency range 303–400 GHz, which is the nominal frequency region accessible with the SHM design. Interestingly, we were able to detect even higher frequencies using second and third mixer harmonics reaching the 600–800 GHz range (second harmonic) and the 1.2 THz region (third harmonic). An attempt was made to detect the DFG signal around 1.6 THz but both the amplitude of the signal and the sensitivity of the mixer at higher harmonics prevented any such detection. The PSD normalized by the intensity of the beatnote (see panel (a) of Fig. 5) shows barely any broadening of the photomixed THz emission, even at the highest detected frequency (1.2 THz). All measurements appear very similar to the results obtained at 1.965 GHz: a narrow peak, with 20 dBz/Hz contrast, settling on top of a broader

pedestal which is most probably the signature of the synthesizer phase noise, scaling with the overall multiplication factor $M$ as $20 \log M$, where $M$ is the chain multiplication factor. As previously, the width of the peaks, 1 Hz at −3 dB, appears to be Fourier transform limited (by the one-second acquisition time) for all frequencies studied, indicating that the short-term FSR fluctuations are close or smaller than what was anticipated. Finally, the phase noise of the THz emission was measured using two photomixers, optimized at 300 and 600 GHz. The first one was a UTC-PD photomixer, waveguide-coupled, with −0.7 V bias voltage and 4.1 mA current, emitting a THz radiation with close to −13 dBm power. The 600 GHz photomixer was an InGaAs photodiode emitter, with −20 dBm emission power. This THz signal was then focused into a THz receiver using a set of two lenses. The THz receiver is composed of a 300 GHz SHM. The first test was done at 300 GHz, and this receiver performed a mixing of the 24th harmonic of the synthesizer frequency (multiplication of the 12.4 GHz by 12, and multiplication by 2 in the SHM, i.e. $M = 24$) that down-converts the 300 GHz signal to 2.4 GHz (intermediate frequency, IF). This IF at SHM output was amplified and fed a phase noise measurement system (E5052B, Agilent). In the experiment, the local oscillator was obtained from a synthesizer with low-phase noise (SMA100B-B711, Rohde & Schwarz). The combination of the available output power from the UTC-PD and the receiver conversion losses induces a noise-floor of the measurement at 300 GHz close to −118 dBc/Hz. At 600 GHz, the signal was measured using the same receiver, operated in mixing the incoming THz signal with the 48th harmonic of the reference source (multiplication of the 12.4 GHz by 12 and multiplication by 4 in the SHM, i.e. $M = 48$). In this case, not only were the conversion losses of the receiver higher, but also the power of the THz source was limited. This resulted in a reduced signal-to-noise ratio (SNR) at the IF output, and consequently a higher noise floor on the phase noise measurement, which is −76 dBc/Hz when combining the second THz source and the receiver operated at 600 GHz. The phase noise measurement

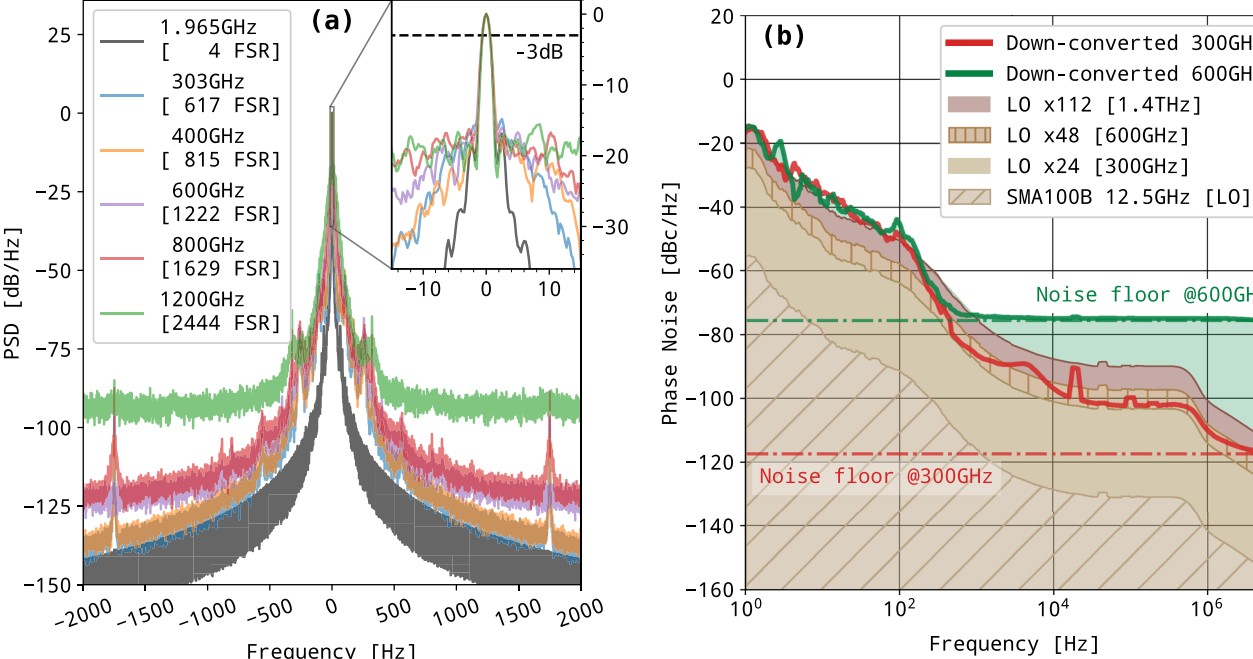

**Fig. 5 | Source noise characterization in the THz domain. a** PSD of the beating signals measured for different THz frequencies. The 1.965 GHz trace corresponds to the previous recordings (Fig. 3). The 600 and 800 GHz traces were recorded using the second harmonic of the SHM. The 1.2 THz frequency was recorded using the third one. **b** Phase noise measurements at 300 and 600 GHz (red and green curve) and their respective experimental noise floor (dotted line). The phase noise

contribution due to the local oscillator (LO) at 12.5 GHz and its increasing calculated contribution at 300, 600, and 1400 GHz (24th, 48th, and 112th harmonics) are plotted in brown. The difference frequency phase noise at 300 and 600 GHz (thick red and green traces) are very similar, only limited by the increasing detector noise (green and red dashed curves).

results are presented in Fig. 5, panel (b). The reference local oscillator (LO), and the phase noise of the 24th and 48th harmonics are shown. At 300 GHz, the performance of the THz source is very close to that of the LO multiplied by the factor $M$. For offsets in the ranges 150–400 Hz and 250–500 kHz, the measured phase noise is limited by the multiplied LO. The measurement at 600 GHz led to a similar phase noise performance, as could be expected from the DFG source. Some variations are observed for a few Hz offsets, but their origin is not identified yet. However, the phase noise reaches a floor value (−76 dBc/Hz) which is due to the limited SNR available at this frequency with the present setup. At the time of the experiments, no efficient THz source/receiver combination was available beyond 600 GHz. Looking further, using this optical source at THz frequencies, for example at 1.4 THz for spectroscopy applications, would require mixing with the 112th harmonic of the LO. The corresponding phase noise is also shown in the figure and, as can be seen, in that case, the optical source leads to a similar phase noise compared to the electronic reference.

### Application to spectroscopy

In the previous section, we have demonstrated that simultaneous optical feedback locking onto a very stable cavity allows common phase noise to be drastically suppressed and leads to the DFG of an ultra-narrow and widely tunable THz radiation. To benchmark the source performance in the context of high-resolution molecular spectroscopy and to confirm the high agility of our source and its frequency purity, we set up three experiments demonstrating (a) source frequency modulated and (b) heterodyne spectroscopy (Doppler in b1 and sub-Doppler in b2). The setups are presented in Fig. 6.

### Source frequency modulated spectroscopy

A series of absorption lines of methanol ($CH_3OH$) vapor were recorded in a range extending from 292 to 800 GHz using the setup depicted in panel (a) of Fig. 6. Coarse frequency tuning was obtained, within one minute and without need for optical realignment, by hopping the laser

from mode to mode of the V-shaped cavity by adjusting the laser chip temperature. Arbitrarily fine continuous tuning was obtained with the OSSBM permitting RF tuning by shifting the THz radiation from 2 to 20 GHz. The photomixing antenna emission beam was collimated with a Teflon lens and coupled to a 2 m long glass cell filled with a few μbar of methanol vapor, at room temperature. The transmitted signal was focused using a second lens onto a Schottky detector (QOD−Zero Bias Detector from VDI). A 48 kHz frequency modulation with 360 kHz depth was applied via the RF fine-tuning source, allowing lock-in detection at the second harmonic of the transmitted signal. The laser emission frequencies were measured using a high-resolution Fizeau wavemeter (WS7 from Highfinesse) with 5 MHz precision. As an example, the absorption signal of the $J_K A/E = 6_{-1}A \leftarrow 5_{-1}A$ methanol line, recorded at room temperature, with a step resolution of 20 kHz and 1 s per spectral point, is shown in Fig. 7. It shows an SNR of about 100 dominated by an oscillating structure that we attribute to stationary waves that develop between the THz optics and slowly evolve during the scan. The second derivative Voigt profile fit of the data yields the expected thermal broadening at 300K (Doppler width), and pressure broadening (Lorentzian width), and a line position in agreement with the parameters given in the HITRAN database[45].

### Heterodyne spectroscopy

To take advantage of the DFG approach we have implemented a detection with an optical receiving antenna, as shown in panel (b) of Fig. 6. We used both a photomixing emitter (TX1, PCA-FD-1550-100-TX-1 from Toptica) and a photomixing receiver (Rx, PCA-FD-1550-130-RX-1 from Toptica). The Rx was fed by Out1 to generate an internal THz wave that interfered with the incident THz wave and gave rise to a corresponding electrical signal. In a homodyne scheme, the THz waves have the same frequency and strong static interference patterns are visible in the spectra. Such patterns can be partially canceled by randomly modulating the THz wave phase, typically mechanically[46]. Here, we took advantage of the high spectral purity of our DFG to set up a

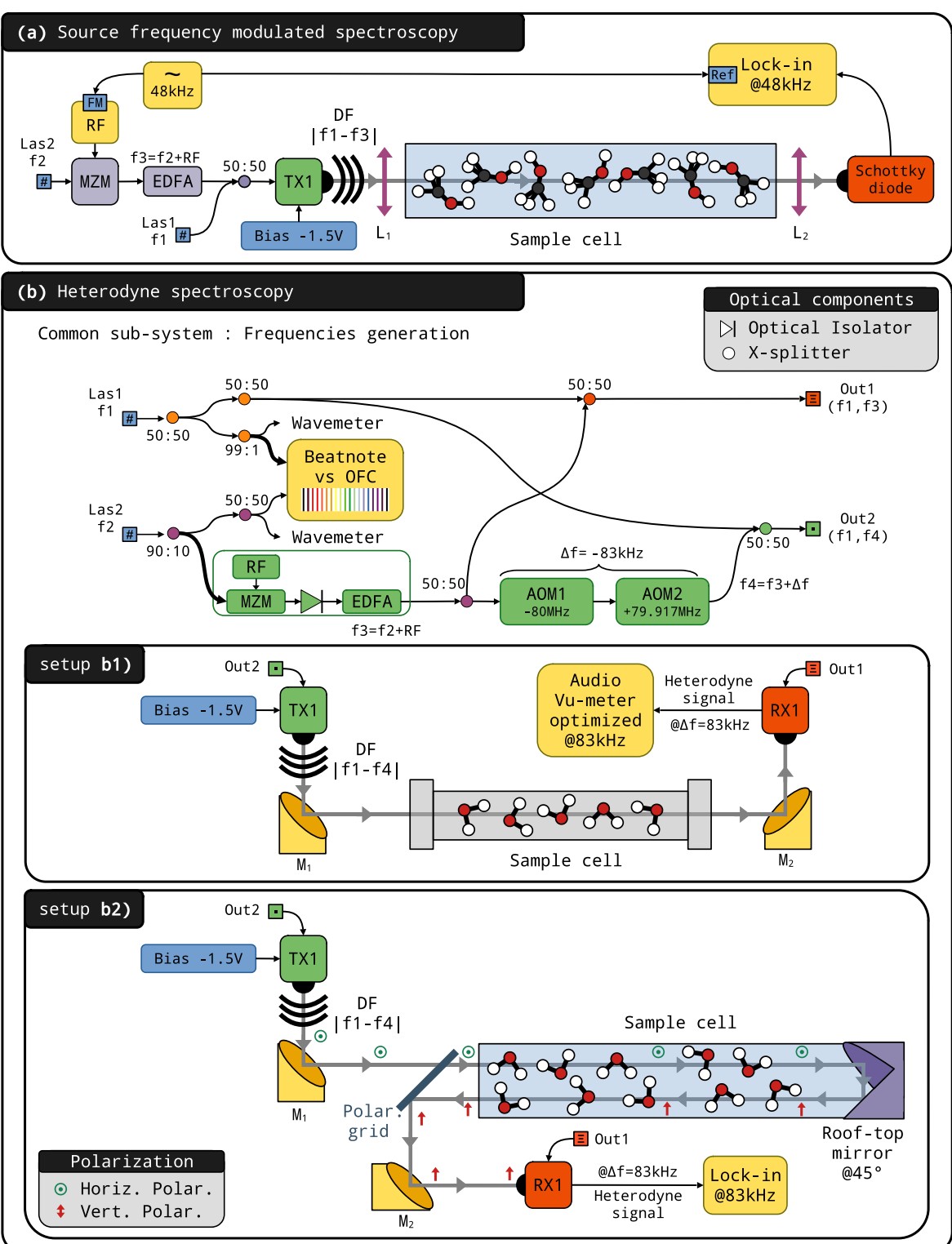

**Fig. 6 | Setups used for spectroscopy, based on Las1 and Las2 beam lines described in Fig. 1. a** A frequency modulation is applied to the RF source (RF). Laser frequencies were monitored using a wavemeter. The photomixing antenna (TX1) emission beam was collimated by a lens (L1, $f = 100$ mm) and sent through a 2 m long glass cell filled with the gas sample at low pressure. The transmitted beam was focused using a second lens (L2, $f = 100$ mm) onto a Schottky detector. The signal was recorded using lock-in detection at the 2nd harmonic of the modulation frequency providing the 2nd derivative profiles of the molecular transitions. **b** The absolute frequencies of both lasers are monitored using a wavemeter and an OFC beatnote. An 83 kHz shifted THz frequency (Out2) is prepared from the initial THz frequency (Out1) using a pair of AOMs (AOM1,AOM2). Panel **b1**: Doppler-limited setup. The emitted (from TX1) and received (on RX) THz beams were collimated and focused using 50 mm off-axis gold mirrors (M1, M2). The single-pass absorption cell was one meter long. The 83 kHz heterodyne signal detected by RX1 was measured using a band-pass power detector (Vu-meter). Panel **b2**: Sub-Doppler setup. The linearly polarized collimated beam passes through a polarization grid, propagates through a double-pass 2 m long absorption cell made of glass, and is then reflected by a roof-top mirror, tilted by 45° from the vertical axis, that rotates the polarization of the beam by 90°. The counter-propagating beam is thus reflected by the polarization grid and detected (by RX1) as a heterodyne signal demodulated by a lock-in amplifier. The molecules experience both propagative and counter-propagative beams, which permits Lamb-dips features to be observed.

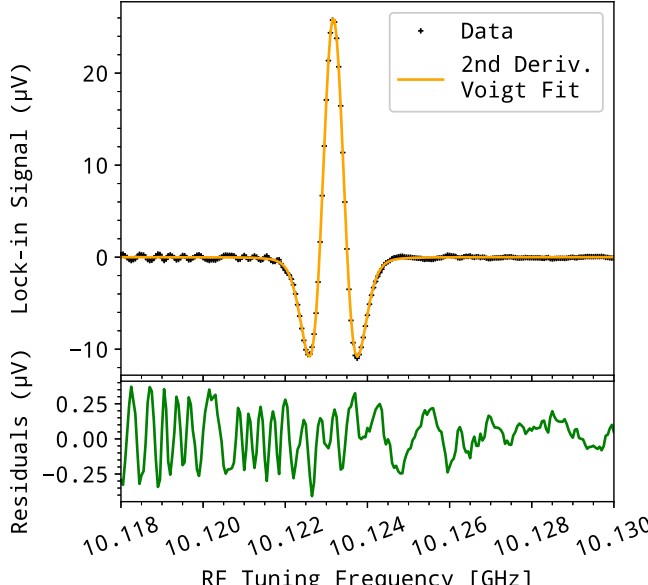

**Fig. 7 | Methanol line recorded by frequency modulation.** Second derivative Voigt fit (orange trace) and residuals (green line) of the $(6_{-1}A \leftarrow 5_{-1}A)$ methanol line (black points) lying around 292.673 GHz. Effective Gaussian and Lorentzian widths are found to be 716 and 233 kHz, respectively. The wavemeter-precision-limited line position retrieved is 292.674(5) GHz, which is in agreement with the HITRAN value of 292.6729(3) GHz.

heterodyne detection scheme thus solving the static interference issue. The Tx frequency Out2 was detuned by 83 kHz with respect to the Rx frequency Out1. As depicted in Fig. 6, this shift was obtained using a pair of acousto-optic modulators (AOM) working on order −1 and +1, and driven at 80 and 79.917 MHz, respectively. The 83 kHz component detected on Rx was amplified, bandpass filtered and the power in this frequency band was retrieved using a full-wave rectifier circuit followed by a narrow bandpass filter stage and an integrator (similar to an audio Vu-meter).

We used a one-meter-long absorption cell to record a series of water absorption lines, in the 300–1400 GHz region, as displayed in panel (b1) of Fig. 6. The water vapor pressure, monitored with a pressure transducer (Baratron 626,MKS), was actively regulated at a pressure of the order of 10 µbar using a closed-loop regulation acting on an inlet solenoid-valve. During the recordings, both laser emission frequencies were accurately determined using both Fizeau wavemeter and their beatnotes with the GPS-referenced OFC. This provides THz frequency determination with sub-kHz level accuracy[47]. Three recorded lines, fitted with a Voigt profile, are shown in Fig. 8 together with their fit residuals. The lines are essentially Doppler broadened from 1.5 to 3.3 MHz. Line positions, once the 83 kHz frequency shift on TX1 is taken into account, are in good agreement with positions listed in the HITRAN database[45]. It is worth noting that jumping to any frequency from 100 to 1400 GHz takes only a few seconds without the need for optical realignment, making the setup particularly suitable for systematic molecular spectroscopy measurements with absolute frequency calibration. Let us emphasize that the heterodyne detection scheme generates a signal that is proportional to the power of the THz beam received. The spectra are therefore recorded in terms of transmission, which gives access to quantitative measurements of line strength. We attribute non-flat residuals to the saturation of the molecular transition under the effect of the THz wave itself[48].

## Sub-Doppler spectroscopy
To further evaluate the potential of our setup for ultra-high resolution molecular spectroscopy measurements, we adapted it to observe

Lamb-dips which are unambiguous signatures of saturation effects. In this third series of spectroscopic measurements, the hyperfine structure (HFS) of the $J_{K_a,K_c} = 1_{1,0} - 1_{0,1}$ water line around 556.936 GHz was targeted. Although unresolved in the Doppler regime at room temperature, it has already been observed by saturation spectroscopy at very low pressure using a frequency multiplication chain[9]. We used a 2 m long glass cell equipped with a rooftop mirror at one end to reflect the THz beam while rotating its polarization plane by $\pi/2$. This arrangement allowed us to separate the reflected THz beam from the incident beam using a polarization grid, as illustrated in panel (b2) of Fig. 6. No attempt was made to precisely control the water vapor pressure which was lower than 1 µbar. To further increase the SNR, the heterodyne detector (the rectifier and its filtering electronics) was replaced by a lock-in detector referenced to the 83 kHz acousto-optics difference frequency. Nine successive scans (160 s each) were recorded with 4 kHz frequency steps and averaged together. The resulting spectrum is presented in Fig. 9. The Lamb-dips, seen here as positive peaks because the raw data are transmission spectra, correspond to the different components of the HFS together with the expected crossovers between the closely spaced components. The overall width of the pattern is 140 kHz and the four expected components of the HFS are visible and well resolved, the fourth being close to the baseline noise. Each component has a width of about 10 kHz and their positions correspond to the literature values[9] within 6 kHz, confirming the accuracy of the THz frequency determination based on the absolute determination of the NIR laser frequencies.

To sum up, we have shown that optical feedback provides an innovative way to stabilize simultaneously the frequency of two NIR lasers down to the Hz level. By photomixing such a pair of telecom lasers stabilized by optical feedback onto a common optical cavity, we have observed THz emission with sub-Hz level linewidth, up to 1.45 THz. We have evaluated the phase noise of the source together with mid- and short-term frequency stability by measuring either the beatnote directly in the THz domain using a frequency multiplication chain or in the NIR region using an OFC. We have demonstrated the suitability of the source for high-resolution spectroscopy by recording methanol and water absorption lines using various detection schemes and confirmed that the transition frequencies could be determined at the kHz level. If it is clear that the agility and versatility of the source, both in terms of coverage and spectral resolution, allow for wide-range spectroscopic recordings, it is also very likely that its very low phase noise will pave the way for innovative THz telecommunications applications.

## Methods
### Phase noise measurement
The phase-noise at 1.965 GHz was directly evaluated using a spectrum analyser. In general, up to 1.2 THz, it was also measured after downconverting the frequency with a frequency mixer. The photo-mixed signal was applied to the RF input of the mixer, and the local oscillator (LO) was connected to the output of a very low noise synthesizer or to the output of a frequency multiplication chain. The phase noise was determined either from the fast Fourier transform (FFT) of the IF signal using a acquisition card with 1 MS/s or with a spectrum analyser. The LO frequency was adjusted in order to obtain an intermediate frequency (IF) around 100 kHz or 2.4 GHz, for FFT- or spectrum analyserbased measurements.

### Long-term drift measurement
The long-term average photo-mixed frequency drift was determined by recording the photo-mixed frequency obtained at 1.965 GHz and complemented by monitoring the absolute frequency of one of the lasers using a high-precision wavemeter and a GPS-referenced optical frequency comb.

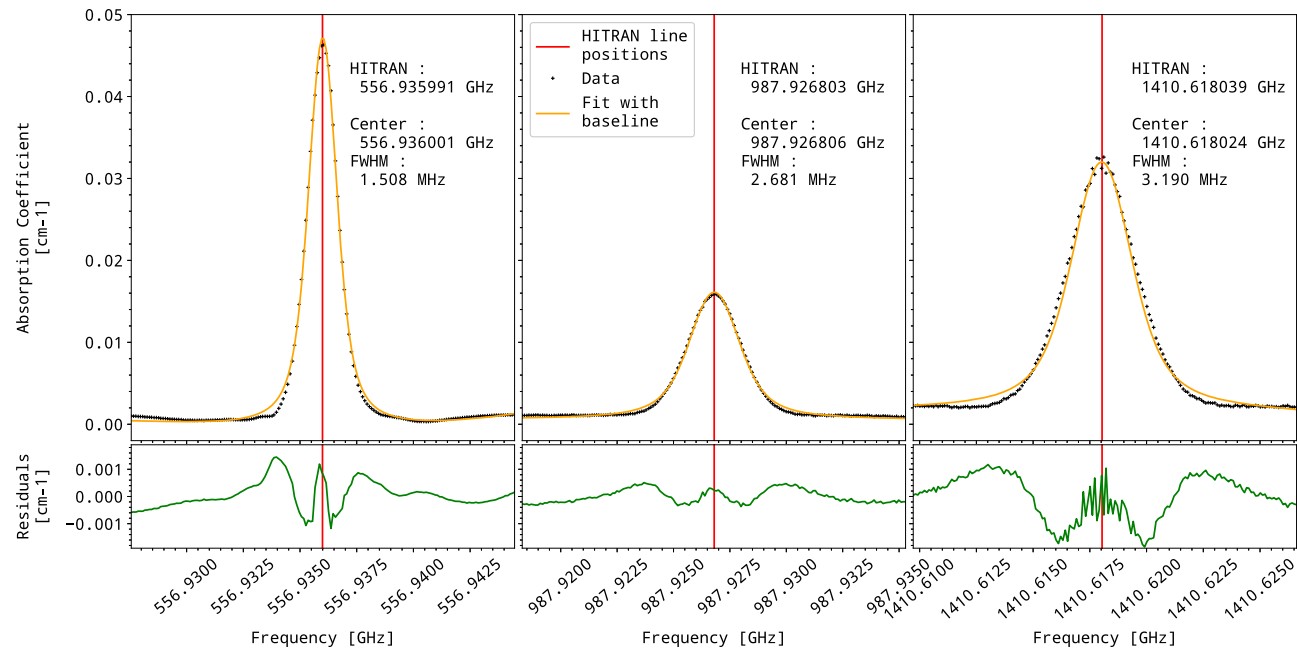

**Fig. 8 | Water lines recorded by heterodyne spectroscopy.** Examples of recorded water lines (black points) around 557, 988, and 1410 GHz were corrected from the 83 kHz frequency shift due to AOM modulators. Fitted positions (orange lines), using a Voigt profile on a third-order polynomial baseline, are compared to the HITRAN database (red sticks). Non-flat residuals (green lines) may suggest the presence of saturation effects (see text).

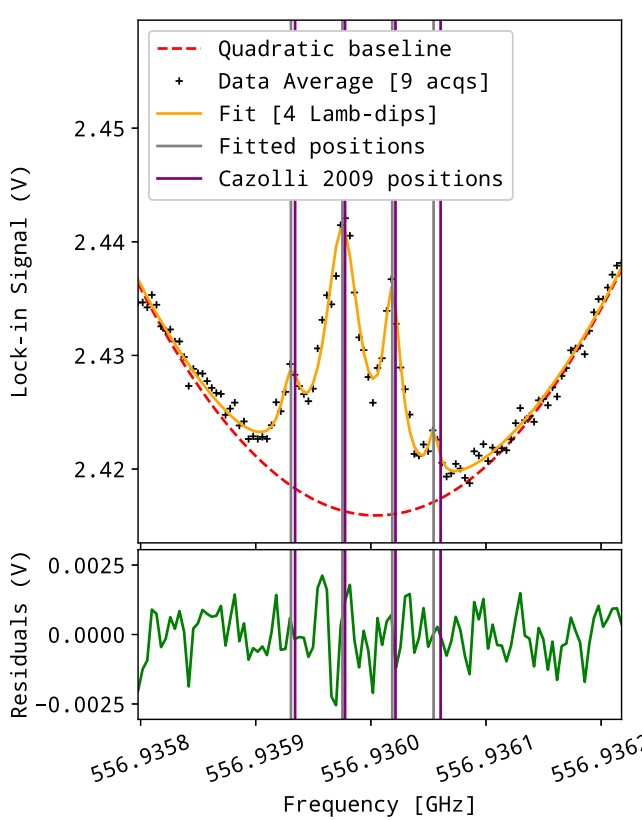

**Fig. 9 | Sub-doppler structure of the water line at 556.935991 GHz.** Four Lamb-dips components were fitted at 556.9359306 GHz (FWHM: 10.1 kHz), 556.9359753 GHz (FWHM: 18.9 kHz), 556.9360186 GHz (FWHM: 9.4 kHz) and 556.9360545 GHz (FWHM: 5.5 kHz), these positions are represented as gray lines. Purple lines are measured positions from the literature[9].

## Spectra acquisition

The spectra were acquired (1) by frequency modulation of the photomixed source through the RF driver of the optical-single-side-band-modulator; a lock-in detection chain was then set-up to detect the second harmonics of the signal detected on a Schottky diode, or (2) by heterodyne spectroscopy made possible by generating two different THz signals, with 83 kHz difference frequency; this difference frequency was filtered out and amplified from the signal received by a photodetector (RX) and a signal directly proportional to the transmission of the cell was obtained.

## Data availability

Data sets generated during the current study are available from the corresponding author on request.

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

## Acknowledgements

We acknowledge financial support from the Agence Nationale de la Recherche (METEORIT, ANR-22-CE30-0036). S.K. acknowledges additional funding from the Agence Nationale de la Recherche (Equipex REFIMEVE+, ANR-11-EQPX-0039). L.L. and S.K. acknowledge funding from the Centre National de la Recherche Scientifique (CNRS, INP pre-maturation program). L.D. and S.K. acknowledge funding from Linksium Grenoble Alpes (project number: 210019). L.D., S.K., and L.L. warmly thank David Terrier, from the mechanical workshop, for the production

of critical parts of the spectrometer in a particularly short time. M.M. and O.P. acknowledge "Investissements d'Avenir" LabEx PALM (ANR-10-LABX-0039-PALM) which supported a part of this work. G.D. and R.K. also state that this work was partially supported by the IEMN Sigmacom platform, the IEMN Flagship on Ultra-High data rates (UHD). The THz testbed also received support from France 2030 programs, PEPR (Programmes et Equipements Prioritaires pour la Recherche), CPER Wavetech. The PEPR is operated by the Agence Nationale de la Recherche (ANR), under the grants ANR-22-PEEL-0006 (FUNTERA, PEPR 'Electronics'). The Contrat de Plan Etat-Region (CPER) WaveTech is supported by the Ministry of Higher Education and Research, the Hauts-de-France Regional council, the Lille European Metropolis (MEL), the Institute of Physics of the French National Centre for Scientific Research (CNRS) and the European Regional Development Fund (ERDF).

## Author contributions

S.K. initially conceived the THz source. L.D., L.L., and S.K. developed the experimental setup. L.D., L.L. and S.K. performed RF phase noise measurements; G.D. and R.K. carried out THz phase noise measurements. L.D., M.M., O.P. and S.K. carried out the spectroscopy; The manuscript was written by L.D. and S.K. with input and review of all authors; S.K. guided the project.

## Competing interests

The authors declare no competing interests.
