## [Peer Review File · Nature Communications]

REVIEWER COMMENTS

Reviewer #1 (Remarks to the Author):

This paper presents a frequency tunable, Hz linewidth terahertz emitter in the 0.1-1.4 THz frequency range. They lock two commercial DFB lasers onto the resonance frequency modes of a single cavity to lower the linewidth and reduce phase noise. For the optical cavity, they use a three mirror V-shaped resonator that they demonstrated before and published in Ref. 33. They feed the locked laser beams to a commercial photomixer to generate terahertz radiation and use it to measure a few gas spectral lines.

While this is a good engineering work and a well-written paper, I do not find it a good fit to Nature Communications that aims to publish technological breakthroughs, and find it more suitable for a specialized photonics journal. This is because the manuscript lacks novelty and the demonstrated performance is orders of magnitude worse than the most recent developments in the field. Specifically,

1) Lowering linewidth and phase noise of a laser using an optical cavity is not new and has been demonstrated/published before by many groups. This includes the authors of this manuscript, who presented linewidth narrowing of the same type of DFB laser used in this work using the same optical resonator used in this work (Ref. 33).

2) Locking two lasers to the same optical cavity for phase noise and linewidth reduction of the generated terahertz beatnote using the same type of photomixer used in this work has been demonstrated/published before. This includes [Shin, D.C., Kim, B.S., Jang, H., Kim, Y.J. and Kim, S.W., 2023. Photonic comb-rooted synthesis of ultra-stable terahertz frequencies. Nature Communications, 14(1), p.790] which demonstrated orders of magnitude narrower linewidth (less than 2mHz), phase noise (-70 dBc/Hz at 1 Hz) and frequency instability (10–15 at 1-s integration) compared to this manuscript.

3) The demonstrated performance (linewidth, phase noise, frequency instability) and spectral impurity (existence of pedestals) are significantly poorer than the state of the art [Shin, D.C., Kim, B.S., Jang, H., Kim, Y.J. and Kim, S.W., 2023. Photonic comb-rooted synthesis of ultra-stable terahertz frequencies. Nature Communications, 14(1), p.790] while requiring a bulky, alignment-sensitive optical resonator, which is not practical for real-world applications.

Here are some additional comments for the authors:

- The paper claims achieving a low phase noise but there is no measurement of phase noise at terahertz frequencies. Phase noise is only calculated at 1.965 GHz and it is mentioned that the phase noise does not change at terahertz frequencies. The authors should select a few frequencies and measure/plot phase noise at each of those terahertz frequencies.

- Similarly, the authors should measure fractional frequency instability as a function of integration time for different terahertz frequencies.

- The authors say "The DFG technique consists of photomixing [18] two independent NIR lasers in a non-linear medium equipped with an antenna that emits photons at the difference frequency of the lasers". This statement is not correct in general and specifically for the Ref. 18 and the photomixers used by the authors, which DO NOT use a non-linear medium for photomixing.

- Figure 4b shows a poor frequency instability for an optical-cavity-locked system (both drift and temporal instability). The authors should explain what are the potential causes of these large frequency fluctuations.

Reviewer #2 (Remarks to the Author):

The authors report on a widely tunable, extremely narrow linewidth continuous-wave THz system. While locking to a reference cavity is a standard and well known process, the obtained results are outstanding and certainly of great interest to the broader THz community. Particularly the tunability of the system is highly important. I recommend revisions as the presentation of the manuscript must be improved.

Comments:

- 1.) Intro, line 43: "non-linear medium equipped with an antenna" this is confusing/mixing of two physically different processes. Usually one uses EITHER a non-linear medium (LiNbO₃, e.g.) OR a photomixer with an antenna. The latter is a linear device (current~power). The THz power still goes quadratic but absorption is a linear process and not governed by Manley Rowe as opposed to non-linear DFG. That's why photomixers excellently work at low THz frequencies while the conversion efficiency of non-linear DFG goes to zero there.
- 2.) Intro: Some further narrow linewidth concepts should be mentioned that are competitive, e.g. continuous-wave electro-optic combs or similar. They also achieve similar performance parameters, are potentially cheaper and more compact, however, with restricted tunability.
- 3.) Page 2, last sentence: It may well be that there are DFB diodes at 1550 nm that can be tuned by 1 THz (those of the authors can be tuned by ~0.9 THz each or so). As far as I understand ref. 28, there were three DFBs implemented, not two.
- 4.) Line 63 (page 3): "[photomixing with standard lasers is] obviously inadequate for tackling gas phase spectroscopy," true, but for other reasons. At normal pressure all gas lines are several GHz wide. Low pressure gas spectroscopy is what you should mention here. 1 MHz may do in most applications at mbar pressures, maybe not astronomic ones.
- 5.) The reader definitely needs more information about the V-shaped cavity as it seems to be the key element of the setup. It is not sufficient to refer to a previous publication of the authors. The paper should be understandable as is. Ideally, present a picture /sketch/cross section of the cavity explaining its setup/working principle. The inset of Fig 1 is a nice photograph but does not convey any information about its working principle.
- 6.) Fig. 1. Doesn't the MZM generate side bands? You would then have multiple colors in your setup!
- 7.) Fig.1.: The two DFB laser signals are orthogonally polarized. At which place are they transformed to the same polarization? Photomixing requires same polarization of both colors.
- 8.) Line 101: The authors say they can tune the lasers by 2 THz. Given the specified maximum difference frequency of 1543-1557 nm yields only 1.74 THz!
- 9.) Fig.9: The arrows indicating the polarization are wrongly indicated. Though it is clear what you mean, the polarization cannot be parallel to the propagation direction. Please fix
- 10.) Language:
 - Intro, line 21: Do you really mean „furtive“? I think it is pretty clear that 6G+ will have to use THz technology
 - Intro, line 42: "independant"-> „independent“
 - Fig. 1: „Lense“-> „lens“
 - The term DFG is sometimes inappropriately used, e.g. in section 3.2: " DFG frequency", spelled out it would mean "difference frequency generation frequency". What is meant is simply the "beat signal". At several instances, also in other sections, DFG should rather be replaced with "frequency difference" (maybe abbreviated "FD") as no difference frequency is generated (yet).
 - Line 188, p10: "Shottky"-> „Schottky“
 - Caption Fig. 7: „two acousto-optic modulator“-> " two acousto-optic modulators"
 - Conclusion: "pair of telecom laser"-> pair of telecom lasers"

Reviewer #3 (Remarks to the Author):

I have reviewed the manuscript by L. Djevahirdian et al. entitled "THz optical synthesis from a low common phase noise, highly tunable, dual frequency source." This article described and discussed the generation of a 0.1 to 1.4 THz continuously tunable terahertz (THz) region. Photomixing two commercial telecom distributed feedback lasers locked by optical feedback onto a highly stable V-shaped optical cavity achieves this. Furthermore, to determine the linewidth of THz radiation, they used supersaturated absorption spectroscopy to measure the Lamb dips of the water vapor line and concluded that the linewidth near 0.5 THz is 30 kHz. By utilizing full telecom band technology, the authors improved the performance as a terahertz light source over previous report. As a scientific technology, it is difficult to find anything novel in this light source, but as a system, it is a THz light source that is nearly perfect.

I appreciate the author's efforts, but I believe that some changes are required for publication in Nature Communications.

1. It's called "dual frequency source" in the title. I am concerned that the term "dual frequency" may mislead the reader in this case. The text is about DFG sources, not two-wavelength generation in the terahertz band. It needs to be corrected precisely so that it does not mislead.

2. It would be difficult to argue that the current light source outperforms the large, complex devices and expensive light sources described in the introduction. It may be using commercial telecom band equipment, but the optics and systems are complex enough.

3. The paper's structure needs to be significantly revised. Because the diagrams of experimental optics (Fig.1), homodyne heterodyne schematic for phase noise detection (Fig.2), heterodyne detection for supersaturated absorption spectroscopy (Fig.7, 9), and experimental apparatus are spread across four different diagrams, they are extremely difficult to read.

4. The description of the V-shape optical cavity, which is the key technology of this light source, is nearly non-existent. The authors should explain the details thoroughly.

5. There is no description of the generated THz light's output. The text should include a figure of THz output power with frequency as a parameter.

6. In the bandwidth of the DFB laser described in line 100, the terahertz frequency range in difference frequency generation is up to 1.748 THz.

7. The experimental results for the agility of the light source described in the abstract are not convincing.

8. For the Lamb dips measurement, no experimental details on saturation absorption spectroscopy were given. This experimental result is critical evidence for determining the THz linewidth of the developed light source, as claimed by the authors. Furthermore, the authors should thoroughly explain how they arrived at the frequency linewidth of 30 kHz.

9. As already reported in Ref. 17, the measurement of Lamb dips by saturation absorption spectroscopy measured here has already been achieved using a Gunn diode in the THz region. As a result, the claimed narrow linewidth THz light source has a weak advantage and is merely a development combining many telecom band technologies.

As mentioned earlier, the system is excellent, but it can be concluded that this paper is poor in terms of spill-over effects for a journal with a high IF, such as Nature Communications.

Response to referees

Dear Referees,

Thank you again for reviewing our article and giving us the opportunity to answer the questions raised by the reviewers in this revised manuscript. Addressing all the suggestions, questions and remarks, as well as gathering all the data and carrying out the new measurements requested by the reviewers, has taken us some time. We are confident that you will confirm that our manuscript now has an even stronger data set, more coherent figures and improved readability.

Specifically, we included experimental phase noise at 300 and 600 GHz to enable easier comparison with the literature, particularly Nature Communications. For this, we now have 2 additional authors, from IEMN, France, a highly renowned institute for THz measurements.

As requested, we have also worked on the overall organization of the text, and propose a solid outline and corresponding figures. The structure of the document is as follows:

1. Introduction
2. Methods/ figure1: DGG experimental setup
3. Source Spectral performances/ figure 2: Characterization setups
 - 3.1. Phase noise in RF domain/ figure 3: PSD & Phase noise
 - 3.2. Long term stability/ figure 4: Raw data & Allan variance
 - 3.3. Performance in the THz/ figure 5: PSD up to 1.2 THz & Phase noise in up to 600 GHz
4. Application to spectroscopy/ figure 6: Spectroscopy setups
 - 4.1. Source frequency modulated spectroscopy/ figure 7: Methanol line @ 300 GHz
 - 4.2. Heterodyne spectroscopy/ figure 8: Water lines up to 1.4 THz
 - 4.3. Sub Doppler spectroscopy/ figure 9: Lamb dip in a water line @ 557 GHz
5. Conclusion

All corrections are shown in red in the manuscript. Moreover, to emphasize this very first demonstration of Lamb dip measurement in the THz range using a purely photonic system, we have adapted the title of the article to "Frequency stable and low phase noise THz synthesis for precision spectroscopy"

You will find our detailed answers below, point by point:

Reviewer #1 (Remarks to the Author):

This paper presents a frequency tunable, Hz linewidth terahertz emitter in the 0.1-1.4 THz frequency range. They lock two commercial DFB lasers onto the resonance frequency modes of a single cavity to lower the linewidth and reduce phase noise. For the optical cavity, they use a three mirror V-shaped resonator that they demonstrated before and published in Ref. 33. They feed the locked laser beams to a commercial photomixer to generate terahertz radiation and use it to measure a few gas spectral lines. While this is a good engineering work and a well-written paper, I do not find it a good fit to Nature Communications that aims to publish technological breakthroughs, and find it more suitable for a specialized photonics journal. This is because the manuscript lacks novelty and the demonstrated performance is orders of magnitude worse than the most recent developments in the field.

We would first like to point out that the quite impressive paper by Shin et al. had not been published at the time of submission. Nevertheless, we have chosen to mention it in the present manuscript.

Specifically,

1) Lowering linewidth and phase noise of a laser using an optical cavity is not new and has been demonstrated/published before by many groups. This includes the authors of this manuscript, who presented linewidth narrowing of the same type of DFB laser used in this work using the same optical resonator used in this work (Ref. 33).

In the present work, the technological breakthrough we present and apply to high-precision molecular spectroscopy in the THz range is threefold:

- a) The lock of two DFBs lasers on the same optical cavity using a relatively simple optical configuration thanks to the powerful optical feedback mechanism.
- b) The continuous tuning of the THz radiation frequency range which makes the setup very versatile.
- c) Demonstration of purely photonic sub-Doppler molecular spectroscopy, which had never been achieved before.

To further emphasize the importance of our developments in cutting-edge spectroscopic applications, we have changed the title of the article to:

“Frequency stable and low phase noise THz synthesis for precision spectroscopy”

In experimental sciences, which are increasingly dependent on technology, we think it is quite difficult to distinguish between novelty, breakthrough and engineering.

The versatility and tunability of our setup set a new standard for very-high precision and accuracy spectroscopy. This is why, with a striking first demonstration (Lamb Dip) and the very wide range of applications that flow from the results we present (like telecoms, radar...), it seemed obvious to us that we had to reach the wide readership of Nature Communications.

2) Locking two lasers to the same optical cavity for phase noise and linewidth reduction of the generated terahertz beatnote using the same type of photomixer used in this work has been demonstrated/published before. This includes [Shin, D.C., Kim, B.S., Jang, H., Kim, Y.J. and Kim, S.W., 2023. Photonic comb-rooted synthesis of ultra-stable terahertz frequencies. Nature Communications, 14(1), p.790] which demonstrated orders of magnitude narrower linewidth (less than 2mHz), phase noise (-70 dBc/Hz at 1 Hz) and frequency instability (10–15 at 1-s integration) compared to this manuscript.

When we submitted our article (January 23, 2023), we were not aware of the publication of Shin et al.'s article (accepted February 3, 2023). Thanks to the referee's comments, we read this article very carefully and now cite it in the revised manuscript. We take this opportunity to highlight several major differences between the two approaches:

- a) The comb is stabilized on the cavity, which makes it difficult to change laser frequencies, as is necessary to benefit from continuous, precise tuning of the THz signal generated.

- b) No demonstration or solution for continuous wave spectroscopy is proposed.
- c) No continuous tuning of the THz frequency is mentioned.
- d) No solution for high resolution spectroscopy is evocated.
- e) The system is not exactly simple: it involves a (state of the art ultra-stable) optical cavity, but also a comb as well as several lasers and series of PLL.
- f) The frequency noise was not explored below 10 Hz when compared to an external source.

3) *The demonstrated performance (linewidth, phase noise, frequency instability) and spectral impurity (existence of pedestals) are significantly poorer than the state of the art [Shin, D.C., Kim, B.S., Jang, H., Kim, Y.J. and Kim, S.W., 2023. Photonic comb-rooted synthesis of ultra-stable terahertz frequencies. Nature Communications, 14(1), p.790] while requiring a bulky, alignment-sensitive optical resonator, which is not practical for real-world applications.*

Indeed, the performance of a single THz frequency synthesized in Shin's paper is better than that demonstrated here. Although these performances are higher than those of our spectrometer, they do not allow a continuous and agile tuning of the THz radiation, which is the mandatory parameter for molecular spectroscopy applications (application number one in the list of applications proposed in Shin's paper).

- a) In the Shin paper, the 100 MHz freq locked to the cavity leads to 100 MHz frequency steps in the THz range which is totally unsuitable for precision molecular spectroscopy in the gas phase at low pressure.
- b) Given that we demonstrate continuous tunability (as also pointed out by the reviewer 2) over the whole range with a minimum frequency step that could be set arbitrarily low, we believe that the approach developed in our work is beyond the state of the art for broadband precision THz spectroscopy.
- c) Furthermore, with regard to the size of the experimental setup that produces the THz radiation, our setup is actually very simple and not as bulky as the reviewer indicates. In contrast, it seems to us that Shin et al's approach is much more complex.
- d) It's worth noting that our system has been moved several times without the need for realignment, so it's actually quite well suited to real-world applications.
- e) Our THz optical alignment is made once and for all, allowing a complete scan of the THz range without the need for any particular realignment.

Here are some additional comments for the authors:

- The paper claims achieving a low phase noise but there is no measurement of phase noise at terahertz frequencies. Phase noise is only calculated at 1.965 GHz and it is mentioned that the phase noise does not change at terahertz frequencies. The authors should select a few frequencies and measure/plot phase noise at each of those terahertz frequencies.

Indeed, it is clear that such a measurement would have made things clearer and was missing from the initial manuscript. To make the paper more convincing, we have included new phase noise measurements at 300 and 600 GHz, performed using the best sources and detectors available to date. They show, as expected, that phase noise remains unchanged up to 600 GHz.

- Similarly, the authors should measure fractional frequency instability as a function of integration time for different terahertz frequencies.

This information is given in the manuscript: the drift of one of the lasers is shown, and the Allan variance of the difference frequency of the two lasers is reported. As explained in the text, it is indeed proportional to the ratio between the difference frequency and the laser frequency.

- The authors say "The DFG technique consists of photomixing [18] two independant NIR lasers in a non-linear medium equipped with an antenna that emits photons at the difference frequency of the lasers". This statement is not correct in general and specifically for the Ref. 18 and the photomixers used by the authors, which DO NOT use a non-linear medium for photomixing.

To avoid any misunderstanding we simply remove the words “non-linear process”.

- Figure 4b shows a poor frequency instability for an optical-cavity-locked system (both drift and temporal instability). The authors should explain what are the potential causes of these large frequency fluctuations.

Two processes are at work. Firstly, our optical cavity (rather inexpensive and manufactured in 2011) is by no means as stable as the one used in the Shin et al. paper. However, we have already developed a much more stable 3-mirror cavity (articles cited), but it was designed for the 1.4 μ m wavelength and was therefore not compatible with our 1.5 μ m photomixers. The second frequency instability seems to result mainly from our PID phase control loop, which is excessively noisy. To convince ourselves of this, we carried out a quick optimization and succeeded in improving the data shown in figure 4. It should be noted that the few Hz of slow excursion it induces have no impact on spectroscopy at kHz level.

Reviewer #2 (Remarks to the Author):

The authors report on a widely tunable, extremely narrow linewidth continuous-wave THz system. While locking to a reference cavity is a standard and well known process, the obtained results are outstanding and certainly of great interest to the broader THz community. Particularly the tunability of the system is highly important. I recommend revisions as the presentation of the manuscript must be improved.

Comments:

1.) Intro, line 43: “non-linear medium equipped with an antenna” this is confusing/mixing of two physically different processes. Usually one uses EITHER a non-linear medium (LiNbO₃, e.g.) OR a photomixer with an antenna. The latter is a linear device (current~power). The THz power still goes quadratic but absorption is a linear process and not governed by Manley Rowe as opposed to non-linear DFG. That’s why photomixers excellently work at low THz frequencies while the conversion efficiency of non-linear DFG goes to zero there.

Our sentence was indeed unclear (see our answer to the first reviewer’s comment). To avoid any misunderstanding, we changed the text accordingly as explained above.

2.) Intro: Some further narrow linewidth concepts should be mentioned that are competitive, e.g. continuous-wave electro-optic combs or similar. They also achieve similar performance parameters, are potentially cheaper and more compact, however, with restricted tunability.

Electro-optical combs and pulsed THz techniques in general are not ideally suited to high-resolution spectroscopy of low-pressure gases.

3.) Page 2, last sentence: It may well be that there are DFB diodes at 1550 nm that can be tuned by 1 THz (those of the authors can be tuned by ~0.9 THz each or so). As far as I understand ref. 28, there were three DFBs implemented, not two.

Indeed, in Ref 28, the authors used 3 DFBs to reach the maximum difference frequency of 2.75 THz. Our sentence was not clear. In our case we actually used only a pair DFBs allowing to reach a maximum frequency span of 1.75 THz. But the same approach – dynamically switching the laser sources to increase the frequency range - is very simple to implement with our setup which is based on fiber lasers.

4.) Line 63 (page 3): “[photomixing with standard lasers is] obviously inadequate for tackling gas phase spectroscopy,” true, but for other reasons. At normal pressure all gas lines are several GHz wide. Low pressure gas spectroscopy is what you should mention here. 1 MHz may do in most applications at mbar pressures, maybe not astronomical ones.

Thank you for this comment. As our main objectives concern high-precision spectroscopy, we now explicitly mention the low-pressure aspect in the sentence. Please note that low phase noise also improves the system's detectivity (heterodyne approach), even when detecting larger structures.

5.) The reader definitely needs more information about the V-shaped cavity as it seems to be the key element of the setup. It is not sufficient to refer to a previous publication of the authors. The paper should be understandable as is. Ideally, present a picture /sketch/cross section of the cavity explaining its setup/working principle. The inset of Fig 1 is a nice photograph but does not convey any information about its working principle.

You are right. Detailing our designed cavity was not our goal, but it is true that a minimum of information had to be added to improve the paper readability. We have added a brief technical description and a simplified sketch.

6.) Fig. 1. Doesn't the MZM generate side bands? You would then have multiple colors in your setup!

We use an I/Q modulator configured to generate a single side band. But it is true that the extinction ratio of both the carrier and other sideband is not infinite. The contrast is 30 dB and residual frequencies (multiple colors) must therefore be taken into account for quantitative measurements below %.

7.) Fig.1.: The two DFB laser signals are orthogonally polarized. At which place are they transformed to the same polarization? Photomixing requires same polarization of both colors.

Both DFB lasers have the same polarization. The user outputs have therefore the same polarization (aligned on the connector key). The polarization beamsplitter rotates the polarization of only one laser. When the light comes back the rotation is compensated. To make it clearer we have improved both the figure and the caption text of figure 1 : "[...] A fibered polarization combiner permits to cross the polarization of one of the lasers before they are injected into the cavity."

8.) Line 101: The authors say they can tune the lasers by 2 THz. Given the specified maximum difference frequency of 1543-1557 nm yields only 1.74 THz!

The reviewer is perfectly right, we change the sentence accordingly.

9.) Fig.9: The arrows indicating the polarization are wrongly indicated. Though it is clear what you mean, the polarization cannot be parallel to the propagation direction. Please fix

This is absolutely true... especially with a TEM₀₀ which has by definition a transverse electric field! The figure has been corrected accordingly.

10.) Language: -Intro, line 21: Do you really mean „furtive“? I think it is pretty clear that 6G+ will have to use THz technology

We deleted furtive

-Intro, line 42: "independant"-> „independent“

Corrected

-Fig. 1: „Lense“-> „lens“

Corrected

-The term DFG is sometimes inappropriately used, e.g. in section 3.2: " DFG frequency", spelled out it would mean "difference frequency generation frequency". What is meant is simply the "beat signal". At several instances, also in other sections, DFG should rather be replaced with "frequency difference" (maybe abbreviated "FD") as no difference frequency is generated (yet).

Absolutely true, thank you! We used DFG as a generic word, which was an inappropriate shortcut. This was misleading and made the text unclear. We modified the text accordingly in several places.

-Line 188, p10: "Shottky"-> „Schottky“

Corrected

-Caption Fig. 7: „two acousto-optic modulator“->“ two acousto-optic modulators“

Corrected

-Conclusion: “pair of telecom laser“-> pair of telecom lasers“

Corrected

Reviewer #3 (Remarks to the Author):

I have reviewed the manuscript by L. Djevahirdian et al. entitled “THz optical synthesis from a low common phase noise, highly tunable, dual frequency source.” This article described and discussed the generation of a 0.1 to 1.4 THz continuously tunable terahertz (THz) region. Photomixing two commercial telecom distributed feedback lasers locked by optical feedback onto a highly stable V-shaped optical cavity achieves this. Furthermore, to determine the linewidth of THz radiation, they used supersaturated absorption spectroscopy to measure the Lamb dips of the water vapor line and concluded that the linewidth near 0.5 THz is 30 kHz.

Actually, this is the molecular transition linewidth. The resolution is much better than this and each resolved sub-transition is close to 10 kHz.

By utilizing full telecom band technology, the authors improved the performance as a terahertz light source over previous report. As a scientific technology, it is difficult to find anything novel in this light source, but as a system, it is a THz light source that is nearly perfect.

I appreciate the author's efforts, but I believe that some changes are required for publication in Nature Communications.

1. It's called "dual frequency source" in the title. I am concerned that the term "dual frequency" may mislead the reader in this case. The text is about DFG sources, not two-wavelength generation in the terahertz band. It needs to be corrected precisely so that it does not mislead.

This is true. We have changed the title accordingly with no reference to misleading “dual frequency” term. We now focus it on spectroscopy:

“Frequency stable and low phase noise THz synthesis for precision spectroscopy”

2. It would be difficult to argue that the current light source outperforms the large, complex devices and expensive light sources described in the introduction. It may be using commercial telecom band equipment, but the optics and systems are complex enough.

In fact, we have demonstrated that the performance of our current system is close to that of multiplication chains (the current reference) and potentially better at higher frequencies according our actualized measurements (figure 5) showing phase noise at 300 and 600 GHz. This is part of the decisive technical breakthrough that has made possible this particularly demanding first demonstration of sub-Doppler THz spectroscopy with a purely photonic THz source. We hope to convince the reader with this new version of the manuscript.

3. The paper's structure needs to be significantly revised. Because the diagrams of experimental optics (Fig.1), homodyne heterodyne schematic for phase noise detection (Fig.2), heterodyne detection for supersaturated absorption spectroscopy (Fig.7, 9), and experimental apparatus are spread across four different diagrams, they are extremely difficult to read.

This was true. Following this remark, we have reorganized the figures and the text structure, as explained above.

4. The description of the V-shape optical cavity, which is the key technology of this light source, is nearly non-existent. The authors should explain the details thoroughly.

We addressed this point by giving a brief description in the text and adding the V-shaped resonator structure in the figure.

5. There is no description of the generated THz light's output. The text should include a figure of THz output power with frequency as a parameter.

Unfortunately, we don't have any power detectors calibrated over the full THz range we operate. We therefore rely on the photomixer data sheet supplied by Toptica.

6. In the bandwidth of the DFB laser described in line 100, the terahertz frequency range in difference frequency generation is up to 1.748 THz.

Absolutely true. It has been corrected.

7. The experimental results for the agility of the light source described in the abstract are not convincing.

To support the agility of the source, we wrote "It is worth noting that jumping to any frequency in the 100GHz to 1400GHz takes only a few seconds, making the setup particularly suitable for systematic molecular spectroscopy measurements with absolute frequency calibration. »
However we have added more information to further convince the reader.

8. For the Lamb dips measurement, no experimental details on saturation absorption spectroscopy were given. This experimental result is critical evidence for determining the THz linewidth of the developed light source, as claimed by the authors. Furthermore, the authors should thoroughly explain how they arrived at the frequency linewidth of 30 kHz.

The referee might be confused on the use of "linewidth" terminology. In the molecular spectroscopy context, the linewidth corresponds to the width of the molecular absorption line. It is about 30 kHz for the Lamb-dip measurements, corresponding to a convolution of the hyperfine structure, the pressure broadening and the transit time of the molecules in the THz radiation field.

The spectral purity of the THz source is Hz-level as demonstrated in the paper, i.e., much below the 30 kHz value.

9. As already reported in Ref. 17, the measurement of Lamb dips by saturation absorption spectroscopy measured here has already been achieved using a Gunn diode in the THz region. As a result, the claimed narrow linewidth THz light source has a weak advantage and is merely a development combining many telecom band technologies.

In Gunn-diode molecular spectroscopy experiments the users have access to a relatively narrow band spectral regions (Ka, E, .. bands). To change band, a tedious realignment effort is required. In contrast, our setup allows us to record Lamb dip spectra both at 300 GHz and almost 1 THz without any adjustment of the spectrometer. Changing from one transition to another (separated by hundreds of GHz) takes only a few seconds.

As mentioned earlier, the system is excellent, but it can be concluded that this paper is poor in terms of spill-over effects for a journal with a high IF, such as Nature Communications.

Our system paves the way for many THz applications and could be at the root of developments by other teams. With a high portability potential, a rather low cost/performance ratio compared to comb-based configurations, and performance that meets a wide range of needs, we are confident that this article will be widely read by the Nature Communication Com readership and highly cited.

In contrast to most of the previous works dealing with THz frequency generation (and published in Nature-related journals), our work is not only a proof of concept of a new narrow-linewidth THz generation method but we further demonstrate the application of this source in the context of precision molecular spectroscopy. The demonstrated large tunability of the present spectrometer together with its ability to be tuned over arbitrarily small frequency steps render our approach a clear opportunity to metrological grade measurements in the whole THz range. Obviously, the source is suitable for a wide range of applications, well beyond spectroscopy. This is why the Nature Communications appears to be the perfect option.

REVIEWER COMMENTS

Reviewer #1 (Remarks to the Author):

I thank the authors for their response and revisions, which strengthened the quality of the manuscript. This paper certainly presents a high-performance terahertz emitter in terms of linewidth and frequency tunability, which are achieved through locking two commercial DFB lasers onto the resonance frequency modes of a single cavity.

However, there are two aspects of the manuscript that stand out:

1) There is no new physics or technical approach used to develop this terahertz emitter.

2) There is no outstanding performance improvement compared to the state-of-the-art. It is true that this work demonstrates continuous frequency tuning, which is very impressive. However, this continuous-wave frequency tunability compared to kHz-MHz step-wise frequency tunability of other frequency-comb emitters does not enable new capabilities for spectroscopy & communication applications. One has to consider the fact that the detectors/receivers in spectroscopy/communication systems have a non-zero bandwidth (like the harmonic mixers used by the authors). Therefore, the detailed spectral information is resolved by the detector without requiring to tune the emitter's frequency with steps smaller than kHz-MHz. That is why scientists have been routinely performing high spectral resolution terahertz spectroscopy for several decades without requiring continuous frequency tunability with less than kHz-MHz step sizes before.

Given the points above, I believe that the manuscript is weak in terms of novelty and technological advancements compared to the most recent developments in the field. But I differ to the editor to decide if the above criteria meet the requirements of Nature Communications or not.

Reviewer #2 (Remarks to the Author):

I still believe it is an excellent manuscript and excellent results, but I agree also with some of reviewer 1's comments. In the end, it is rather an editorial decision on whether or not to publish with Nature Communications. I would like to emphasize that state-of-the-art gas spectroscopy setups use purely electronic systems. There do exist photonic ones, but so far, they are not really competitive. The authors have demonstrated with their manuscript that they can outperform typical stabilities and resolutions of electronic systems using a photonic approach which is also a major selling point. Generally speaking, photonic approaches become more and more versatile and competitive with classic approaches and this is one of the papers proofing this.

Besides, I only have three minor comments:

1.) I disagree with the answer on my 2nd comment: "Electro-optical combs and pulsed THz techniques in general are not ideally suited to high-resolution spectroscopy of low-pressure gases." First of all, because EO combs can be CW and be much more robust and simpler to use than your method. They may, however, have some issues with tuning range (though they are simply tunable by the EO frequency!). Second, there are comb techniques that are specifically made for spectroscopy with a tuning range and tuning speed exceeding yours, see e.g.:

<https://ieeexplore.ieee.org/abstract/document/8874007>

<https://www.nature.com/articles/s41598-020-59398-1>

The second reference is on the optical subsystem only. Although the resolution of this specific system is on the order of 10 kHz and thus worse than yours, it is excellently suited for low pressure spectroscopy as the linewidth is already on the order of the Doppler linewidth. Also, typical spectroscopic applications frequently do not reach the Doppler-broadened linewidth as they operate at several Pa pressure level.

2.) General question to the MZM: To my experience, the MZM RF generator has a linewidth and stability in the 1 Hz range. If you do fine tuning with the MZM, how can you achieve mHz/s stability as claimed in the abstract? Fig.4 (b) supports the long term stability of 0.17mHz/s but the short term stability taken from Fig. 4 (b) is rather on the order of 10 Hz (amplitude of blue graph around average, short term excursions). For a true experiment, rather the short term excursions matter as you would not be able to resolve a line smaller than these excursions.

3.) For the motivation by the authors of employment in communication applications, I would consider the setup too bulky and too expensive, so I do not see that this is going to happen.

Reviewer #3 (Remarks to the Author):

I reviewed for the second time the manuscript entitled "THz photosynthesis from low common-mode noise, highly variable, dual-frequency sources" by L. Djevahirdian et al. The authors have sincerely responded to and corrected my comments.

I mentioned in the first review that the system is very complete for THz spectroscopy, and the authors have demonstrated that it can be applied to molecular spectroscopy.

Unfortunately, no novel physics or significant scientific breakthroughs can be identified. However, in view of the editorial policy of Nature Communications, I judged that the progress as a system is very significant.

For these reasons, I am accepting this manuscript for publication in Nature Communications.

Response to reviewers

Dear Referees,

Thank you for this new reviewing of our article. Corrections in the introduction and section 3.2 appear in blue. Please find our detailed answers below, point by point:

Reviewer #1 (Remarks to the Author):

I thank the authors for their response and revisions, which strengthened the quality of the manuscript. This paper certainly presents a high-performance terahertz emitter in terms of linewidth and frequency tunability, which are achieved through locking two commercial DFB lasers onto the resonance frequency modes of a single cavity.

However, there are two aspects of the manuscript that stand out:

1) There is no new physics or technical approach used to develop this terahertz emitter.

This terahertz emitter permitted the first proof of high-resolution spectroscopy by photomixing with the demonstration of sub-doppler recordings. This paper is about the development of an instrument setting a new standard for THz production via photomixing. It is sure that new photomixers, reaching higher frequencies will be developed in the near future. Our setup will permit to straightforwardly take advantage of it.

Using this setup, we have already detected a THz spintronic-based emission and presented it in a conference paper (IEEE DOI: [10.1109/IRMMW-THz50927.2022.9896051](https://doi.org/10.1109/IRMMW-THz50927.2022.9896051)). Such a detection was impossible with current technologies.

Therefore, we strongly believe that this paper is the first milestone towards new physics exploration, beyond 1 THz, and as such will be highly read and cited.

2) *There is no outstanding performance improvement compared to the state-of-the-art. It is true that this work demonstrates continuous frequency tuning, which is very impressive. However, this continuous-wave frequency tunability compared to kHz-MHz step-wise frequency tunability of other frequency-comb emitters does not enable new capabilities for spectroscopy & communication applications.*

As mentioned by reviewer 2, in terms of performance the instrument presented in our study outperforms typical stabilities and resolutions of other photonic based systems while reaching frequencies that are not easily accessible using electronic approaches, and it does so with continuous frequency tunability.

We agree that a THz spectrometer with a resolution step ranging from kHz to MHz is generally adequate for classical gas-phase molecular spectroscopy in the low-pressure regime. Indeed, many molecular spectroscopy results in this range have already been obtained over the last few decades. However, when performances in terms of ultimate resolution and frequency metrology are required (for example, using molecular spectra to test the temporal and spatial dependence of fundamental constants of physics), the very fine spectral components, which can be separated by a few kHz in the case of hyperfine splittings, need to be fully resolved and well sampled. An ultra-fine THz source (typically at Hz level) AND a finely tunable frequency step (typically at sub-kHz level) are therefore mandatory, independently of the choice of the detection scheme (see answer below). For these applications, which represent an important field of future molecular physics and require the exhaustive study of a wide variety of molecular targets, we are convinced that our device offers a particularly credible and suitable alternative.

One has to consider the fact that the detectors/receivers in spectroscopy/communication systems have a non-zero bandwidth (like the harmonic mixers used by the authors). Therefore, the detailed spectral information is resolved by the detector without requiring to tune the emitter's frequency with steps smaller than kHz-MHz. That is why scientists have been routinely performing high spectral resolution terahertz spectroscopy for several decades without requiring continuous frequency tunability with less than kHz-MHz step sizes before.

We believe the reviewer suggests transient spectroscopy (pulsed or chirped-pulse, which requires high excitation power) or more generally emission spectroscopy. The non-zero electrical bandwidth of the detector then permits to set up a heterodyne detection scheme where the local oscillator is a THz source with high spectral purity (such as the one published in the article by Shin et al. Nat. Comm 2023 or our own). The intermediate frequency is then finely analyzed using a radio-frequency spectrum analyzer.

Transient-Heterodyne: We hope to be able to implement the transient approach in the near future using our own source coupled to a high bandwidth detector.

Heterodyne: Alternatively, we also plan to set up a heterodyne spectrometer based on a synchrotron source (with our source as the local oscillator) like some of our co-authors have developed (see e.g. Tamaro et al, Nat. Comm 2015, Hearne et al, Optics Express 2022). Observe that even if this is possible, such an instrumental development is far from trivial and is certainly not available to the spectroscopic community for decades. Note that such a heterodyne detection scheme will not enable the saturation spectroscopy required for sub-Doppler precision measurements.

Given the points above, I believe that the manuscript is weak in terms of novelty and technological advancements compared to the most recent developments in the field. But I differ to the editor to decide if the above criterial meets the requirements of Nature Communications or not.

Reviewer #2 (Remarks to the Author):

I still believe it is an excellent manuscript and excellent results, but I agree also with some of reviewer 1's comments. In the end, it is rather an editorial decision on whether or not to publish with Nature Communications. I would like to emphasize that state-of-the-art gas spectroscopy setups use purely electronic systems. There do exist photonic ones, but so far, they are not really competitive. The authors have demonstrated with their manuscript that they can outperform typical stabilities and resolutions of electronic systems using a photonic approach which is also a major selling point. Generally speaking, photonic approaches become more and more versatile and competitive with classic approaches and this is one of the papers proofing this.

Besides, I only have three minor comments:

1.) I disagree with the answer on my 2nd comment: "Electro-optical combs and pulsed THz techniques in general are not ideally suited to high-resolution spectroscopy of low-pressure gases." First of all, because EO combs can be CW and be much more robust and simpler to use than your method. They may, however, have some issues with tuning range (though they are simply tunable by the EO frequency!). Second, there are comb techniques that are specifically made for spectroscopy with a tuning range and tuning speed exceeding yours, see e.g.:

<https://ieeexplore.ieee.org/abstract/document/8874007>

<https://www.nature.com/articles/s41598-020-59398-1>

The second reference is on the optical subsystem only. Although the resolution of this specific system is on the order of 10 kHz and thus worse than yours, it is excellently suited for low pressure spectroscopy as the linewidth is already on the order of the Doppler linewidth. Also, typical spectroscopic applications frequently do not reach the Doppler-broadened linewidth as they operate at several Pa pressure level.

We thank the reviewer for this important comment. This nice work permitting to lock an ECDL onto a virtually shifted comb-tooth using an EOM is now cited in our introduction. Since we are focusing on cw-THz generation, we prefer not to discuss about purely pulsed THz techniques. Corrections appear in blue in the new manuscript.

We fully agree that the set-up of Puppe et al. described in the abstract of IRMMW conference is very pertinent to regular molecular spectroscopy studies where Doppler and pressure broadening lead to MHz molecular line widths in the THz range. But it remains quite a complex system to implement and requires two expensive ECDLs in addition to a very stable optical frequency comb, ideally itself locked against a reference laser locked to an ultra-stable cavity (as in Shin et al. Nat. Comm 2023) to improve the source linewidth.

High-precision measurements for a large variety of molecular samples require ultimate spectral purity and metrology of the THz source and we believe our source is a good alternative setup.

2.) General question to the MZM: To my experience, the MZM RF generator has a linewidth and stability in the 1 Hz range. If you do fine tuning with the MZM, how can you achieve mHz/s stability as claimed in the abstract?

We have not observed any stability issue with the MZM yet. The RF generator we use (R&S SMA 100B) is stated for 0.053nHz internal resolution and sub-Hz linewidth at 20 GHz. Let's emphasize that we do not adjust dynamically the MZM control voltages, which may be the source of the linewidth you observe? But such an effect might still be hidden in the noise excess explain below.

Fig.4 (b) supports the long term stability of 0.17mHz/s but the short term stability taken from Fig. 4 (b) is rather on the order of 10 Hz (amplitude of blue graph around average, short term excursions). For a true experiment, rather the short term excursions matter as you would not be able to resolve a line smaller than these excursions.

It is true that the phase control error signal of the current system suffers from artifacts due to etaloning. We are currently working at improving that point that we hope will be solved in a near future. The long

term trend, however, indicates the potential for the technique and confirms that the physics behind it is well understood. We had excursion of a few Hz but that must be compared to existing photomixing continuous-wave devices which are well outperformed. Regardless, in the sake of transparency, we have added a statement in the text discussing the current limits on the short-term stability in section 3.2.

3.) *For the motivation by the authors of employment in communication applications, I would consider the setup too bulky and too expensive, so I do not see that this is going to happen.*

We agree that there is still a huge amount of work to be done in various fields, such as integration and miniaturization, before we can achieve everyday communication applications, and that this will not be ready in the near future. However, we hope that current performance levels, and their potential for improvement, will initiate developments in this direction.

Reviewer #3 (Remarks to the Author):

I reviewed for the second time the manuscript entitled "THz photosynthesis from low common-mode noise, highly variable, dual-frequency sources" by L. Djevahirdian et al. The authors have sincerely responded to and corrected my comments. I mentioned in the first review that the system is very complete for THz spectroscopy, and the authors have demonstrated that it can be applied to molecular spectroscopy.

Unfortunately, no novel physics or significant scientific breakthroughs can be identified. However, in view of the editorial policy of Nature Communications, I judged that the progress as a system is very significant. For these reasons, I am accepting this manuscript for publication in Nature Communications.

Thank you for this positioning.

We hope that the reviewers and editor will be convinced of the value of this paper, and that the performance and potential demonstrated will inspire the Nature Communication reading community, far beyond molecular spectroscopy.

REVIEWERS' COMMENTS

Reviewer #2 (Remarks to the Author):

Although I do not believe that the setup can be successfully miniaturized and it would therefore not be a sensible choice for communication systems, I still believe it is a very decent report and there were hardly any Lamb-dip resolved measurements presented in the literature. The accuracy of the system is excellent. I agree to some degree with reviewer 1 that the individual methods were already present but I am not aware of a published system with similar performance features. I consider it very significant. I therefore recommend acceptance at Nature Communications.

After reading the manuscript again, I found some minor issues that should still be corrected:

Intro: "amplitude and phase modulation is only possible prior to frequency multiplication" is not generally true. There are several papers on (sub-)THz modulators. Some of them you can just put into the optical beam, others are waveguide-coupled. One simple example is a pin diode switch in a WRxx waveguide. I have also read papers about graphene-based modulators.

Line 49: A remark on the term "Difference frequency generation": It is usually reserved for non-linear processes ($\chi^{(2)}$ materials) and is commonly NOT used for photomixing, which is a linear absorption process, despite there are many similarities.

Remarks on chapter 3.2/Fig.4: I still think the claim "sub-Hz stability" although there are very obvious 10 Hz excursions on a short time scale is not valid. Please don't confuse drifts (which are indeed few mHz/s) and stability (short term fluctuations).

Comment on author's response on telecom applications of their system and on other comb systems that would be too expensive (no change on the manuscript necessary): 1.) Telecom: As you need a stable, UHV cavity, it is pretty clear that communication applications will not be possible in a realistic scenario. No one would use a UHV system with pumps. Available comb systems: As your system requires vacuum and temperature stabilization, it is way bulkier and more difficult to operate than a comb laser, locked, e.g., to a GPS clock. The latter comes for free.

Line 274: "Let us emphasize that the heterodyne detection scheme generates a signal that is proportional to the power of the THz beam received." Is power really correct? Shouldn't it read "field"? Homodyne and heterodyne detection schemes are field and phase-sensitive

References, line 447: please replace the "?" by " μ "

Response to reviewers

Dear Referees,

Thank you for this new reviewing of our article. Please find our detailed answers below, point by point:

Reviewer #1 (Remarks to the Author):

REVIEWERS' COMMENTS

Reviewer #2 (Remarks to the Author):

Although I do not believe that the setup can be successfully miniaturized and it would therefore not be a sensible choice for communication systems, I still believe it is a very decent report and there were hardly any Lamb-dip resolved measurements presented in the literature. The accuracy of the system is excellent. I agree to some degree with reviewer 1 that the individual methods were already present but I am not aware of a published system with similar performance features. I consider it very significant. I therefore recommend acceptance at Nature Communications.

After reading the manuscript again, I found some minor issues that should still be corrected:

Intro: “amplitude and phase modulation is only possible prior to frequency multiplication” is not generally true. There are several papers on (sub-)THz modulators. Some of them you can just put into the optical beam, others are waveguide-coupled. One simple example is a pin diode switch in a WRxx waveguide. I have also read papers about graphene-based modulators.

We agree that it is fundamentally possible to modulate phase and intensity in the THz domain, but it remains much more complex than doing it prior to multiplication and much easier to do so in the optical domain. In consequence, “only” has been replaced by “essentially” to temper the statement.

Line 49: A remark on the term "Difference frequency generation": It is usually reserved for non-linear processes (chi(2) materials) and is commonly NOT used for photomixing, which is a linear absorption process, despite there are many similarities.

As the frequency generated is equal to the difference of the two laser frequencies it was difficult to strictly avoid the "difference frequency generation" term in the text. But we used 18 times the terms photomixer, photomixing, photomixed.

Remarks on chapter 3.2/Fig.4: I still think the claim "sub-Hz stability" although there are very obvious 10 Hz excursions on a short time scale is not valid. Please don't confuse drifts (which are indeed few mHz/s) and stability (short term fluctuations).

The frequency excursion remains sub-Hz (RMS) for up to 10 seconds as shown in the Allan deviation plot.

Comment on author's response on telecom applications of their system and on other comb systems that would be too expensive (no change on the manuscript necessary): 1.) Telecom: As you need a stable, UHV cavity, it is pretty clear that communication applications will not be possible in a realistic scenario. Noone would use a UHV system with pumps. Available comb systems: As your system requires vacuum and temperature stabilization, it is way bulkier and more difficult to operate than a comb laser, locked, e.g., to a GPS clock. The latter comes for free.

Line 274: "Let us emphasize that the heterodyne detection scheme generates a signal that is proportional to the power of the THz beam received." Is power really correct? Shouldn't it read "field"? Homodyne and heterodyne detection schemes are field and phase-sensitive

Let us remind that the heterodyne detection scheme generates an output signal that is proportional to the power of the received THz beam, while keeping amplitude and phase information thanks to the linear behavior of the mixer, ie the IF output power is linear to the incoming THz power at mixer input.

References, line 447: please replace the "?" my " μ "

Corrected.

Thank you again for your deep review of the manuscript.